# The carbon budget of the managed grasslands of Great Britain - informed by earth observations

Vasileios Myrgiotis[1], Thomas Luke Smallman[1], and Mathew Williams[1]

[1]School of GeoSciences and National Centre for Earth Observation, University of Edinburgh, Edinburgh EH9 3FF, UK

**Correspondence:** Vasileios Myrgiotis (v.myrgiotis@ed.ac.uk)

**Abstract.**

Grasslands cover around two thirds of the agricultural land area of Great Britain (GB) and are important reservoirs of organic carbon (C). Direct assessments of the C balance of grasslands require continuous monitoring of C pools and fluxes, which is only possible at a small number of experimental sites. By relying on our quantitative understanding of ecosystem C biogeochemistry we develop models of grassland C dynamics and use them to estimate grassland C balance at various scales. Model-based estimation of the C budget of individual fields and across large domains is made complex by the spatial and temporal variability in climate and soil conditions, and in livestock grazing, grass cutting and other management activities. In this context, earth observations (EO) provide sub-field resolution proxy data on the state of grassland canopies allowing us to infer information about vegetation management, to apply observational constraints to the simulated ecosystems and, thus, to mitigate the effects of model input data uncertainty. Here, we show the potential of model-data fusion (MDF) methods to provide robust analyses of C dynamics in managed grasslands across GB. We combine EO data and biogeochemical modelling by implementing a probabilistic MDF algorithm to (1) assimilate leaf area index (LAI) times series (Sentinel-2), (2) infer defoliation instances (grazing, cutting) and (3) simulate livestock grazing, grass cutting, and C allocation and C exchanges with the atmosphere. The algorithm uses the inferred information on grazing and cutting to drive the model's C removals-and-returns module according to which $\approx$1/3 of C in grazed biomass returns to the soil as manure (other inputs of manure not considered) and C in cut grass is removed from the system (downstream C emissions not considered). Spatial information on soil C stocks is obtained from the SoilGrids dataset. The MDF algorithm was applied for 2017-2018 to generate probabilistic estimates of C pools and fluxes at 1855 fields sampled from across GB. The algorithm was able to effectively assimilate the Sentinel-2 based LAI time-series (overlap=80%, RMSE=1.1 $m^{-2}m^{-2}$, bias=0.35 $m^2m^{-2}$) and predict livestock densities per area that correspond with independent agricultural census-based data ($r$=0.68, RMSE=0.45 LU $ha^{-1}$, bias=-0.06 LU $ha^{-1}$). The mean total removed biomass across all simulated fields was 6 ($\pm$1.8) tDMha$^{-1}$y$^{-1}$. The simulated grassland ecosystems were on average C sinks in 2017 and 2018; the net biome exchange (NBE) was -191$\pm$81 (2017) and -49$\pm$69 gCm$^{-2}$y$^{-1}$ (2018). Our results show that the 2018 European summer drought reduced the strength of C sinks in GB grasslands and led to a 9-fold increase in the number fields that were annual C sources (NBE>0) in 2018 (18% of fields) compared to 2017 (2% of fields). The field-scale analysis showed that management in the form of timing, intensity and type of defoliation were key determinants of the C balance of managed grasslands, with cut fields acting as weaker C sinks compared to grazed fields. Nevertheless, extreme weather, such as prolonged droughts, can convert grassland C sinks to sources.

**Abbreviations**

GPP: Gross Primary Productivity

$R_a$: Autotrophic Respiration

$R_h$: Heterotrophic Respiration

$B_g$: Grazed Biomass

$B_c$: Cut Biomass

GCD: $B_g - B_c$

M: Manure produced by grazing livestock

Reco: Ecosystem Respiration (Reco = $R_a$ + $R_h$ )

NEE: Net Ecosystem Exchange (NEE = Reco - GPP)

NBE: Net Biome Exchange (NBE = NEE + $B_c$ + $B_g$ - M)

NPP: Net Primary Production (NPP = GPP - $R_a$)

LAI: Leaf Area Index

CI: Confidence Interval

RCR: Relative Confidence Range ($100 \times CI \div$ mean)

SD: Standard Deviation

SOC: Soil Organic Carbon

$\Delta_{SOC}$: Change in SOC pool size

AGB: Aboveground Biomass

LU: Livestock Units

VPD: Vapour Pressure Deficit (Pa)

## 1 Introduction

Grasslands, natural and managed, are important biomes globally, with large soil carbon (C) pools and a key role in the cycling of water and nutrients (Ostle et al., 2009). In Great Britain (GB), approximately two thirds of the agricultural land is grassland, managed at varying intensities as part of livestock farming systems (DEFRA, 2020). According to their biomass productivity and management intensity GB grasslands are grouped into rough grazing (low productivity), permanent (medium productivity) and temporary (high productivity) grasslands (Qi et al., 2017). The environmental impacts of grassland management increase with its intensity. Impacts range from local-scale air and water pollution, due to manure production and nutrient loss, to emissions of all three major global warming-causing greenhouse gases (GHG) i.e. $CO_2$, $CH_4$ and $N_2O$ (Herrero et al., 2016; Vertès et al., 2018). Quantifying how C travels through the coupled system of the atmosphere, grass, livestock and soil is challenging, due to their dynamic management (grazing, cutting, re-seeding, fertiliser and manure application) (Felber et al., 2016; Fetzel et al., 2017; Conant et al., 2017; Blanke et al., 2018; Abdalla et al., 2018). However, quantifying C cycling in grasslands is a prerequisite for shaping, implementing and monitoring policies for reducing the climate impact of managed grasslands in GB and across the world (Committee on Climate Change, 2019; Sollenberger et al., 2019).

Grasses fix C through photosynthesis (gross primary production, GPP), and allocate a fraction of this C to grow stems, leaves and roots. Plant senescence results in transfers of biomass C to litter and dead organic matter in the soil which undergo decomposition. Defoliation, through grazing and cutting, is a major disturbance to C cycling (Gastal and Lemaire, 2015; Skinner and Goslee, 2016). Grass cutting abruptly removes most of the aboveground C from the ecosystem, forcing the grass

to rebuild the leaf area necessary for photosynthesis and growth. In contrast, livestock grazing causes frequent but less intense removals of aboveground biomass C. A fraction of the grazed C accumulates in livestock biomass but most of it exits the animal's body as manure, respiration-$CO_2$ and digestion-$CH_4$. The amount of grazing-based manure C that is added to the soil's dead organic matter pool varies significantly depending on farm-level manure management decisions (Dangal et al., 2020).

The potential of managed grasslands in GB, and beyond, to act as C sinks (Mcsherry and Ritchie, 2013; Ward et al., 2016; Chang et al., 2017; Abdalla et al., 2018; Pawlok et al., 2018) is premised on achieving a negative C balance at the ecosystem scale. Net ecosystem exchange (NEE) quantifies the C balance at the ecosystem scale based purely on gas fluxes and is equal to the difference between ecosystem respiration (Reco = $R_h$ + $R_a$) and GPP. Net biome exchange (NBE) quantifies the C balance including lateral flows connected to cutting and animal management. NBE is equal to NEE after accounting for removals (grazing, cutting) and inflows (manure deposition) of C to the ecosystem. Animal-based C fluxes to the atmosphere (respiration $CO_2$, digestion $CH_4$) and other ecosystem-scale C losses (leached C, manure-induced $CH_4$) are sometimes included in the calculation of NBE (Soussana et al., 2007). NEE makes up the bulk of NBE and is measured at field-scale using closed chambers and eddy covariance towers with both techniques having contrasting strengths and weaknesses, and requiring expert knowledge to deploy (Riederer et al., 2014, 2015). NBE calculation requires measurements of the lateral flows on which human management plays a major role (Chang et al., 2021).

Quantitative understanding of the dynamics of C pools and fluxes in grasslands is gained through field and lab-based experiments. This understanding is incorporated into models of ecosystem C biogeochemistry, which are conceptually-coherent structures of mathematical equations that track the fluxes of C in the atmosphere-plant-soil-livestock system (Snow et al., 2014; Chang et al., 2013; Ma et al., 2015; Sándor et al., 2018; Puche et al., 2019). Biogeochemical models can upscale knowledge on ecosystem C dynamics across large areas and over time. Model-based upscaling represents a robust way for diagnosing the role of climate and management on C exchanges, and exploring C sensitivity of future climate and alternate management scenarios. Models require information on environmental conditions as inputs. Providing these inputs across space introduces uncertainty (input uncertainty) to model predictions because the relevant data come from spatial extrapolations of point measurements (i.e. soil surveys, weather stations). Another key source of input uncertainty is the lack of accurate spatial data on grassland management i.e. harvest and grazing patterns and manure and fertiliser use, which must therefore be inferred by some means (Vuichard et al., 2007; Chang et al., 2015a; Fetzel et al., 2017; Rolinski et al., 2018; Blanke et al., 2018; Abdalla et al., 2018; Chang et al., 2021). Model credibility can be supported by effective calibration with ground data and validating predictions using independent data. Providing uncertainty estimates on model outputs provides robust contexts for model interpretation (Kennedy and O'Hagan, 2001; Dietze, 2017).

Advances in satellite-based remote sensing methods, i.e. Earth Observation (EO), over the past decade have increased the volume and resolution of spatial data on grassland states (e.g. sward biomass, chlorophyll content) and soil factors (e.g. soil moisture and temperature) (Reinermann and Asam, 2020; Ustin and Middleton, 2021; Zeng et al., 2022). There is an opportunity to use EO data in model-based studies to perform upscaling with reduced input and parametric uncertainty and with constrained predictive uncertainty and model bias. In this context, high resolution (<100 m), frequently-retrieved (∼weekly)

EO data on the state of grassland vegetation can be assimilated and used to validate relevant model predictions and to calibrate model parameters at the scale of individual grassland fields (Patenaude et al., 2008; Oijen et al., 2011; Maselli et al., 2013; Pique et al., 2020b, a). In addition, time-series of EO-based vegetation indices can be used to monitor vegetation volume change and identify the timing of the relevant management i.e. grass harvesting and livestock grazing (Dusseux et al., 2014; Giménez et al., 2017; Yu et al., 2018; Reichstein et al., 2019). Leaf area index (LAI) conveys information on vegetation structure and volume, and can be estimated from multispectral optical EO data (Munier et al., 2018). The volume of EO-derived LAI data is, however, dependant on the frequency of satellite overpasses and the level of cloudiness.

In previous analyses at two grassland eddy flux sites in GB we have shown that calibrating biogeochemical model parameters with ground-based LAI observations allowed robust diagnoses of the effects of grazing and cutting on independently measured net C exchanges (Myrgiotis et al., 2020). In a follow-up study at another grassland research farm in GB we demonstrated that model calibration with satellite-based LAI observations was effective for monitoring biomass removals and quantifying management impacts on field-scale C balance (Myrgiotis et al., 2021). Here, we build on this earlier work to demonstrate how EO data and biogeochemical modelling can be combined to (1) detect defoliation instances (i.e. grass cutting and grazing intensity) and (2) to estimate the variation in C dynamics over a large domain (GB) and at fine resolution (sub-field scale). We use a parsimonious process-based biogeochemical model of grassland C dynamics (DALEC-Grass) that is integrated into a probabilistic model-data fusion (MDF) algorithm (CARDAMOM). DALEC-Grass is driven by weather data and field-specific EO-based data on weekly change in vegetation volume. CARDAMOM performs field-specific calibrations of DALEC-Grass parameters by assimilating local EO LAI time-series. ]The MDF algorithm is implemented for 2017-2018 on a large sample of 1855 managed grassland fields in GB (England, Wales and Scotland). We obtain a sample representative of the different grassland ecosystems and management types by randomly selecting 1 field per 25 km$^2$ across GB from the Land Cover Map (vector land parcels) of the United Kingdom (UK). Grazing intensity, cutting timing and yields and C pools and fluxes are predicted by DALEC-Grass for every simulated field. In order to evaluate our MDF analysis, we compare predictions of annual grass yields (grazed and cut biomass) to biomass utilisation data from the relevant literature and to livestock density data from the most recent GB agricultural census data. In addition, in 2018 GB was affected by a summer heat and drought wave (summer 2018 was $\approx 1^o$C warmer than summer 2017) allowing us to examine the impact of climate anomalies on grassland C balance (Kendon et al., 2018, 2019). The aim of this study is to answer four key questions:

1. Can we detect realistic variations in grassland vegetation management over national domains at field scale by assimilating EO information on LAI?

2. What is the C balance of managed grasslands and how does it vary across GB?

3. Which factors control the predicted C balance and biomass removals?

4. How large is the analytical uncertainty on C cycling and which factors affect it?

The novelty of this research is to combine EO data and modelling to infer management of grasslands at field-scale across a nation and then to simulate the role of management on grassland C exchanges. The advent of highly-resolved satellite data

from Sentinel 2 makes this possible, allowing tracking of ∼weekly changes in LAI at sub-field scales for a national sample
of grassland fields. The intermediate complexity model employed means that Bayesian approaches to model calibration can
explore the uncertainty of parameters and estimates of C cycling. The key innovation is to combine observations of changes
in grassland LAI from space with expected changes in LAI (i.e. grass growth rates) derived from process modelling. The
difference between observed and expected change in LAI is used to infer consumption by grazing livestock or removals by
grass cutting. The C cycle estimate is then updated based on this estimation.

## 2  Materials and methods

### 2.1  Materials

#### 2.1.1  Location of managed grasslands

For the identification of the location and limits of representative grassland fields we used the 2018 Land Cover Map plus (LCM),
which is updated annually by the Centre of Ecology and Hydrology (CEH) of the UK (www.ceh.ac.uk/crops2015). The LCM
includes geo-referenced polygons of improved grassland fields in GB that are identified as such by using a combination of
reflectance data. The LCM data are validated against ground observations of land-use type.

#### 2.1.2  DALEC-Grass Model

DALEC-Grass (Fig. 1) is a process-orientated model of intermediate complexity representing C cycling in grassland ecosys-
tems. DALEC-Grass uses meteorological information to calculate GPP, autotrophic and heterotrophic respiration, changes in
LAI, the C turnover of different plant and soil pools as well as the removal of C via grazing and grass cutting. Photosynthesis
is calculated using the Aggregated Canopy Model (ACM) and phenology is calculated using the Growing Season Index (GSI)
approach (Williams et al., 1997; Smallman et al., 2017). DALEC-Grass uses a dynamic scheme to allocate C to above and
below-ground plant tissues, which is based on the assumption that C allocation to roots increases after sufficient leaf area has
been developed such that there are diminishing returns on further canopy expansion (Myrgiotis et al., 2020). The model uses a
simple scheme to describe C allocation to soil with two pools considered; a more labile litter pool to which dead plant material
and manure-C are added, and a more recalcitrant soil organic carbon pool (SOC), which receives C from the litter pool only. At
each time-step the model uses EO-based input data on change in vegetation volume. This information is translated into grazing
or cutting instances (see section 2.2.2). The simulated grazed biomass-C is converted to animal respiration $CO_2$-C, digestion
$CH_4$-C and manure-C using generic conversion factors (see Fig. 1) extracted from the relevant literature (Parsons et al., 2009;
Zeeman et al., 2010; Worrall and Clay, 2012; Bell et al., 2016; Lee et al., 2017). These generic conversion factors are used
because the type, weight and age of animals grazing on individual fields can neither be inferred from EO data nor be reliably
estimated from available datasets of livestock spatial distribution (e.g. agricultural census). All the manure that the simulated
livestock produce is added to the litter pool. The conversion factors used in DALEC-Grass reflect an averaging of relevant data
for beef and dairy cattle and sheep, which are the main types of livestock in the GB. When harvest is simulated the harvested

biomass-C is removed from the ecosystem. The model's 31 parameters are presented in Table A1 in Appendix A. In this study, DALEC-Grass is implemented at a weekly time-step.

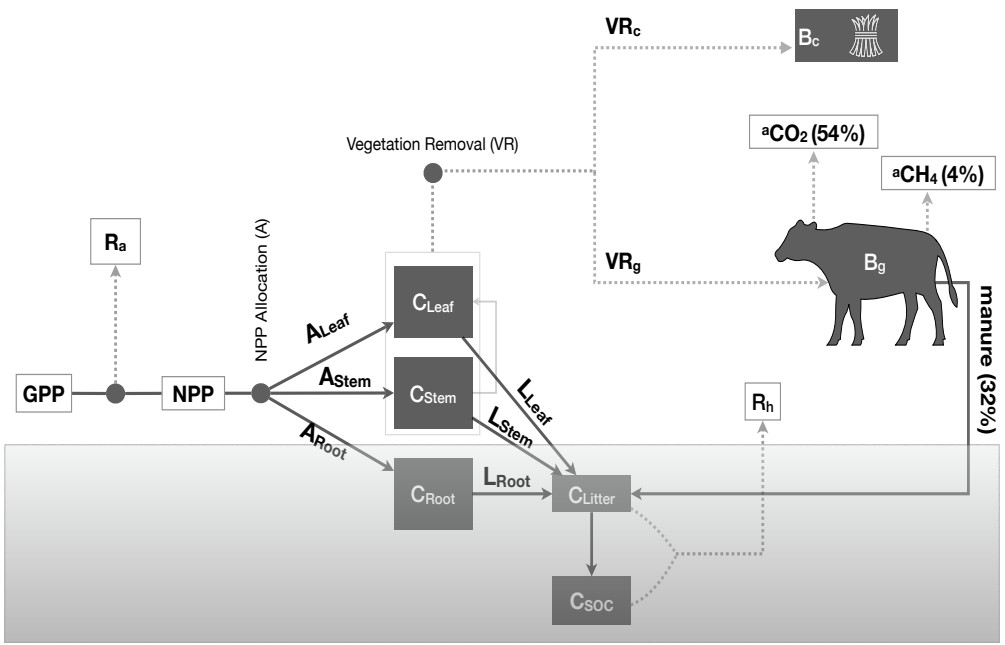

**Figure 1.** Schematic description of the DALEC-Grass model. DALEC-Grass simulates the dynamics of 5 C pools (C) : leaf, stem, roots, litter and SOC. C is allocated to the 5 C pools via NPP allocation (A) and litter production (L). Vegetation removals (VR) can occur due to grazing or cutting. DALEC-Grass determines whether a vegetation removal is caused by grazing, cutting or neither (see section 2.2.2). When cutting is simulated (VR$_c$>0) cut biomass (B$_c$) is removed from the ecosystem. When grazing is simulated (VR$_g$>0) 32% of grazed biomass (B$_g$) is converted to manure, 54% of grazed biomass (B$_g$) is converted to animal respiration ($^a$CO$_2$) and 4% of grazed biomass (B$_g$) is converted to methane ($^a$CH$_4$). Dotted lines ($\cdots$) show outward fluxes of C. Solid lines (—) show internal and inward fluxes of C

### 2.1.3 Carbon Data Model Framework (CARDAMOM)

The CARbon DAta MOdel FraMework (CARDAMOM) is a Bayesian MDF framework that is tailored for use in ecosystem biogeochemistry studies (Bloom et al., 2016). By assimilating observational data CARDAMOM updates the distribution of model parameters following the rules of Bayesian inference. A key aspect of CARDAMOM is the use of ecological and dynamic constraints (EDC), which are conditions applied to the parameter sampling process in order to ensure the mathematical, ecological and biogeochemical sensibility (or "common sense") of the simulated system (Bloom and Williams, 2015). In simple terms, CARDAMOM examines whether the simulated pools and fluxes that result from implementing the model with a sampled parameter vector behave in realistic ways; i.e. do not exceed certain user-defined, widely-accepted and literature-based limits. The EDCs used in CARDAMOM are presented in Table A2 in Appendix A. A schematic description of how CARDAMOM and DALEC-Grass are connected is provided in Figure A1 in Appendix A.

Bayesian inference is performed in CARDAMOM using the root mean square error (RMSE) between the simulated and the EO-based LAI time-series to calculate and attribute likelihoods to every sampled parameter vector. In this study, the Simulated Annealing (SA) algorithm is used to implement the probabilistic parameter sampling process (Kan et al., 2016). After testing the SA algorithm to identify the optimal number of repetitions for achieving acceptable chain convergence the number of parameter proposals was set to 5,000,000. The uncertainty around the assimilated LAI data was set to 15% (relative standard deviation) in this study. However, it should be noted that the uncertainty around remote sensing-based LAI data is poorly determined but largely underestimated (Zhao et al., 2020).

A uniform distribution was used for each of the 31 DALEC-Grass parameter priors and the range for each parameter prior is presented in Table A1 (Appendix A). In Myrgiotis et al. (2020) DALEC-Grass parameter priors were refined through implementing the model using known vegetation management (cutting dates, livestock density time-series) and by assimilating field-measured LAI and $CO_2$ flux data (chamber-based and eddy covariance). In Myrgiotis et al. (2021) these model priors have been tested and refined further using EO-based vegetation reduction time-series (as vegetation management-related model drivers) and by assimilating Sentinel-2 (S2) based LAI time-series. The limits of the uniform parameter distributions used in the present study are based on the results of Myrgiotis et al. (2021) but 4 parameters were allowed to vary more than these results suggested in order to better consider the variability in management factor across GB grasslands (indicated with * in Table A1 in Appendix A). These parameters are the plant photosynthetic N use efficiency (PNUE), the leaf mass C per area (LCA), and the pre-grazing and pre-cutting biomass.

### 2.1.4 Earth observation data

Two independent EO datasets on leaf area index (LAI) were used in this study. The first dataset is the Copernicus Global Land Surface (CGLS) LAI data product. CGLS LAI data comprise top-of-atmosphere reflectance products from the Proba-V satellite processed into LAI. The CGLS LAI data have a spatial resolution of 300m and a temporal resolution of 10 days. Gaps in the CGLS LAI time-series due to cloud coverage are filled using a machine learning model built with time-series for past years and neighbouring pixels (Smets et al., 2018). For each simulated field the corresponding time-series has been converted, from their original 10-day time-step, to a weekly time step using linear interpolation. Thereafter, the reduction between subsequent dates in the time-series was calculated. When the change between week $n$ and $n+1$ was positive the reduction value for week $n$ is 0. Hereafter, we refer to this time-series as the "vegetation reduction" time-series. We note that the vegetation reduction time-series are input drivers of DALEC-Grass that inform its C removals-and-returns module (described in section 2.1.2).

The EO-based LAI data that are assimilated in CARDAMOM were calculated from Sentinel-2 (S2) images. Sentinel-2 is an EO mission of the European Space Agency (ESA) that consists of two optical-imaging polar-orbiting satellites that were launched in 2015 (S2A) and 2017 (S2B). Atmospherically-corrected images at 10m, 20m and 60m resolutions (L2A product) were downloaded from the Amazon Web Services (AWS) S2 data pool. All the available cloud-free images that cover GB for 2017-2018 were processed to remove pixels with cloud and haze and, then, used to calculate LAI (at 20m) using the sen2cor algorithm (Weiss and Baret, 2016). When available, the field-average S2-based LAI value that corresponds to the day closest

to the first day of every simulated week is added to the weekly observational LAI time-series that are assimilated through the CARDAMOM MDF framework.

### 2.1.5 Environmental and management data

Six meteorological drivers are used in DALEC-Grass to drive variations in biogeochemical process: (1) minimum and maximum temperature ($^o$C), (2) total short-wave radiation ($MJm^{-2}d^{-1}$), (3) atmospheric $CO_2$ concentration (ppm), (4) 21-day average photoperiod (sec), (5) 21-day average minimum T and (6) 21-day average vapour pressure deficit (Pa). Data were obtained from the ERA5 global atmospheric reanalysis database of the European Centre for Medium-Range Weather Forecasts (ECMWF). Values of soil C ($gCm^{-2}$ at 60cm depth) at 300m resolution were obtained from the the SoilGrids database (Hengl et al., 2017). For every simulated field the mean and standard deviation (SD) of the corresponding SoilGrids pixels are used to define the range of the model's initial SOC pool size parameter. In absence of robust spatial data on manure application in grasslands we do not consider human-controlled application of manure (i.e. manure spreading). All the manure C that the simulated livestock produce after grazing is directly added to the soil's litter pool. This is the only C input to the ecosystem apart from the atmospheric $CO_2$-C assimilated in biomass.

Agricultural census-based data on the number of sheep and cattle (beef and dairy) were obtained from the EDINA AgCensus database (AgCensus, 2020). They are used in this study to independently evaluate the estimates from the MDF implementation. The AgCensus data are produced by spatially disaggregating the numbers of cattle and sheep recorded at the level of local administrative units into a 5km grid. The most recently available livestock data for each constituent country of GB refer to different years; 2010 for England, 2015 for Wales and 2017 for Scotland.

## 2.2 Methodology

### 2.2.1 Sampling of grassland fields from the national land cover map

Implementing the MDF algorithm for the thousands of fields that are classified as improved grassland in the LCM database is computationally demanding and time consuming. In addition, the spatial resolution of the CGLS (Proba-V-based) vegetation reduction data is 300m (9 ha). Taking into account that the average managed grassland field is 5-9 ha in area, we set a minimum limit of 6 ha (and a maximum of 13 ha) when filtering the LCM dataset to obtain the location of fields. Moreover, the number of EO data points available for each field depends on the time of image capturing and the amount of cloud cover at overpass. As a consequence, the number of dates of available EO data can vary considerably between fields. We set a limit of having at least 30 S2 data points (for 2017-2018) for a field to be selected for simulation. The fields that met the conditions were allocated to 25km$^2$ cells of a 5km grid of GB. One field was randomly selected from each cell, which resulted in a set of 2108 fields (Fig, 2). The CARDAMOM MDF algorithm was implemented for each of the selected fields for 2017 and 2018 by running DALEC-Grass at a weekly-time step while assimilating the corresponding available EO-based LAI data. We refer to the outputs of this implementation as "MDF predictions".

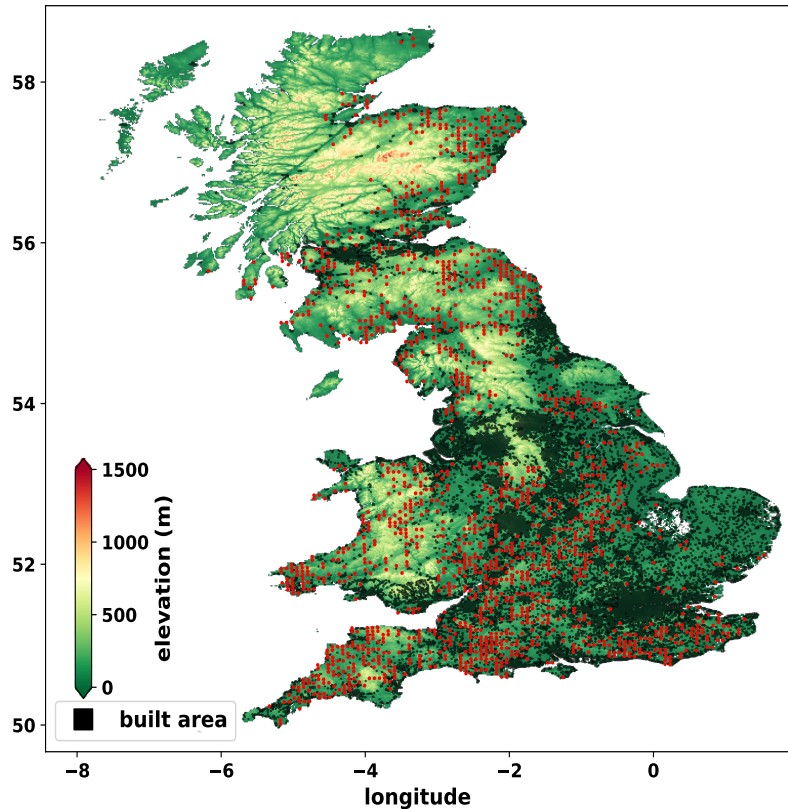

**Figure 2.** Topographic map of Great Britain (GB) with red symbols showing the locations of sampled fields. Built-up areas are shown in black. Digital elevation model from Pope, A. (2017).

### 2.2.2 Identifying and calibrating grazing and cutting from EO data

We used the two independent EO datasets on LAI to inform the analysis. The first dataset (CGLS) was used to estimate weekly absolute LAI change (vegetation reduction time-series), and based on the magnitude of change, to derive whether cutting or grazing had occurred. CGLS data therefore provided an independent estimate of management operations and act as a driver of LAI loss in the model, week-to-week. The value of CGLS is the availability of continuous weekly data, with no gaps in coverage. Their weakness is the large uncertainty on LAI change, particularly for smaller fields (<9ha) where edge effects can introduce substantial biases. The second LAI dataset (S2) was used directly in the assimilation scheme to adjust DALEC-Grass parameters and minimise the mismatch between observed S2-based and simulated LAI. This assimilation adjusts the processes driving LAI dynamics, and hence the LAI losses initially estimated with CGLS data. The outcome is a more robust description of LAI dynamics and management impacts. The strength of S2 data is in the high resolution which ensures within-field accuracy. Their weakness is the frequent gaps in the S2 time-series which leads to some weeks lacking S2 LAI information. It should be noted that gap-filling the S2 LAI time-series using simple interpolation methods is not appropriate in the case of

managed grasslands. This is because grazing and cutting can take place at any time during the growing season. Therefore, assuming that, for instance, grass grew freely (no defoliation) when two S2 LAI data points weeks apart suggest so, would not be correct.

In more detail, DALEC-Grass simulates weekly biomass growth driven by weather. LAI loss is then imposed by CGLS estimates. Broad uncertainty on the CGLS estimates recognise the potential bias in this driving data set (Myrgiotis et al., 2021). At each weekly time step the LAI loss from CGLS is used to decide whether a field has undergone a grass cutting event or a livestock grazing event or neither. For a vegetation reduction data point to be simulated as a cutting: (1) the event should occur between April and October, when cutting tends to occur; (2) the simulated aboveground biomass at the time of cutting should be greater than the pre-cutting biomass parameter (P28, see Table A1) indicating that there is enough grass for a cut to be worthwhile; and (3) the resulting yield should be > 80 g C m$^{-2}$, another economic test for cutting. If any of these 3 conditions is not met then a grazing event is indicated and simulated if the simulated aboveground biomass exceeds the pre-grazing biomass parameter (P27, see Table A1). Otherwise, if neither a cut nor a grazing event can be simulated, no LAI removal is simulated. The processes of inference and simulation of grazing/cutting are performed inside the model and depend directly on certain relevant parameters and indirectly on most parameters, which control the simulation of photosynthesis and allocation in general. Therefore, during the process of assimilating observational S2 LAI the vegetation management-related decisions made by DALEC-Grass are conditioned on observations. This approach links the noisy CGLS vegetation reduction drivers with the constraint on parameters from the more accurate high resolution S2 based observational LAI data (Fig. 3). Assimilation is performed by implementing DALEC-Grass while sampling from the model's parameter space and minimising the error (RMSE) between observational S2-based and simulated LAI time-series (see section 2.1.3).

### 2.2.3 Calculation of presented variables and sign convention

The micrometeorological sign convention is used when presenting C balance variables, whereby a positive (+) sign before a NEE and NBE value signifies an addition of C to the atmosphere, and a negative sign (-) signifies a removal of C from the atmosphere. The net change (gCm$^{-2}$) in the size of the soil organic C (SOC) pool is presented and referred to as $\Delta_{SOC}$. Positive $\Delta_{SOC}$ values signify an increase in the SOC pool size and negative values signify a decrease in the SOC pool size. We use the difference between total annually grazed and cut biomass (GCD) to quantify the relative impact of these two vegetation removals methods. A negative (-) GCD value signifies that more biomass was removed from the simulated field via cutting than it was via livestock grazing. A positive (+) GCD value signifies that more biomass was removed via livestock grazing than it was via grass cutting.

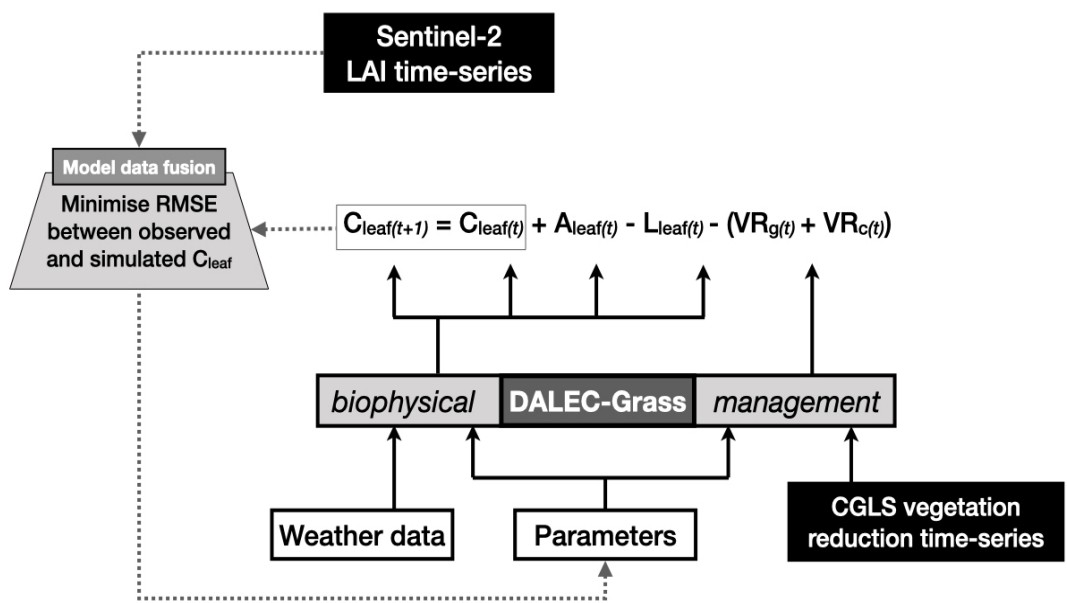

**Figure 3.** Description of how S2 LAI observations, (CGLS) vegetation reduction time-series and DALEC-Grass are used to infer and calculate managed vegetation removals (grazing, cutting). The DALEC-Grass biophysical module simulates weekly leaf growth and senescence driven by weather data. The DALEC-Grass management module simulates weekly vegetation removals driven by the vegetation-reduction data. The CARDAMOM MDF algorithm calibrates the parameters of DALEC-Grass in order to achieve the smallest possible error (RMSE) between S2 LAI and simulated LAI time-series. LAI = $C_{leaf}$ ÷ P15 (see table A1 for details on parameter P15). The DALEC-Grass management module determines whether LAI reduction is due to grazing ($L_g$) or cutting ($L_c$) or neither. When a grazing reduction is identified ($VR_g$>0 thus $VR_c$=0) the livestock C-turnover process (described in 2.1.2) is implemented. When a reduction is identified as produced by cutting ($VR_c$>0 thus $VR_g$=0) most leaf biomass is removed; parameters P27, P28, P29, P31 (table A1) play a direct role in cut yield estimation. When neither grazing nor cutting is identified then $VR_c$=0 and $VR_g$=0. Boxes in black show EO-based information. $C_{leaf}$: leaves C pool, A: C allocation, L: litter production, $VR_g$: vegetation removal due to grazing, $VR_c$: vegetation removal due to cutting, t: time.

$$NEE = Reco - GPP \tag{1}$$

$$Reco = R_a + R_h \tag{2}$$

$$NBE = NEE + B_c + B_g - M \tag{3}$$

$$\Delta_{SOC} = C \text{ flux to SOC} - C \text{ losses from SOC} \tag{4}$$

$$GCD = B_g - B_c \tag{5}$$

where NEE: Net Ecosystem Exchange, NBE: Net Biome Exchange, REco: Ecosystem respiration, Ra: Autotrophic respiration, $R_h$ : Heterotrophic respiration, $B_c$: Cut biomass, $B_g$: Grazed biomass, M: manure, GCD: grazed - cut biomass, $B_g$ : Grazed biomass, $B_c$: Cut biomass. Note that C flows into the SOC pool from the litter pool only and C is lost from the SOC pool via

heterotrophic respiration (Fig. 1). All variables are presented in gC m$^{-2}$ t$^{-1}$, where t is the time period over which the results are summed (e.g. 3-months, year).

### 2.2.4 Assessment and analysis of MDF predictions

The effectiveness of the LAI assimilation process is assessed by quantifying the level of fit between MDF-predicted and EO-based LAI time-series using (1) the % of overlap between the EO-based data points (field mean) and the corresponding
MDF-predicted ranges (95% confidence interval); and (2) the RMSE and (3) the bias between the simulated and observed time-series. To account for the possibility that some of the simulated fields may not be managed grasslands due to changes in management, but classified as such in the LCM data, we remove from the results any fields for which the estimated overlap is < 50% (see results for size of post-MDF dataset).

To answer our first science question, the MDF-predicted weekly grazed biomass is converted into livestock units (LU) per ha
following the assumptions that : (1) 1 cattle is 1 LU and one sheep is 0.11 LU; (2) 1 LU weighs 650kg; (3) an animal demands $\approx 2.5\%$ of its weight in the form grass dry matter (DM) when grazing; and (4) 47.5% of DM consists of C (Vertès et al., 2018). The MDF-predicted and independent census-based LU ha$^{-1}$ are compared using the correlation coefficient ($r$) and the RMSE as the assessment metrics. The MDF-based estimates of grass biomass utilisation across GB are assessed against data from the Qi et al. 2017 study.

To answer our second science question we present and examine the annual and seasonal C balance and the cumulative annual fluxes of the simulated fields. To assess what controls the predicted C balance of the simulated grasslands (our third science question) we quantify the correlation coefficient between meteorological model drivers, management-related model parameters and MDF predictions of C cycling. In order to provide a more quantitative assessment of the factors that control grassland C dynamics we quantify the relative impact of management and climate on the MDF-predicted NBE. We use the model
meteorological drivers and the posterior model parameters related to management and climatic controls for every simulated field to train a random forest (RF) model that estimates NBE. 75% of the data are used to train the RF model and 25% to assess its predictive ability (coefficient of determination). Thereafter, we use the Shapley Additive Explanations (SHAP) method to quantify how much each RF-predictor affects the RF-predicted NBE (Rodríguez-Pérez and Bajorath, 2020). The SHAP method examines the structure of the RF model and provides the weight (SHAP value) that the model gave to each predictor. SHAP
values can be seen as the machine learning equivalent of the coefficient of determination ($r^2$). The estimated SHAP values are normalised (0-1) to be comparable to $r^2$. We note that RF is used in this study solely to support MDF data analysis and not for predictive purposes.

For each simulated field and model output the MDF algorithm produces a mean and 95% confidence interval. To answer our fourth science question, we quantify the predictive uncertainty around an output by calculating its relative confidence
range (RCR). RCR is equal to the size of the MDF-predicted 95% confidence interval divided by the corresponding mean, and expressed as %. We present and examine the estimated RCRs to identify the key factors that affect uncertainty.

## 3 Results

### 3.1 Assimilation of EO-based LAI data

For 12% of the initial dataset (2108 fields) our analysis failed to generate a simulated-vs-observed LAI overlap > 50%. These fields were removed from the analysis and the final reported dataset includes 1855 fields. Based on the 1855 fields, three performance metrics indicated that CARDAMOM effectively assimilated the provided EO-based LAI time-series (Fig. A2 in Appendix). Thus, CARDAMOM could identify parameter values for DALEC-grass for each field so that the model could effectively reproduce the phenological development of the canopy, consistent with meteorological forcing and a realistic removal of grass by grazing and/or cutting. The overlap between EO-based and simulated LAI was 80 ($\pm$11) %, the RMSE was 1.1 ($\pm$0.22) $m^{-2}m^{-2}$ and the bias was 0.35 ($\pm$0.40) $m^{-2}m^{-2}$. There were no clear spatial patterns (Fig. A2) in the error statistics across GB and no obvious geographical biases.

### 3.2 MDF-predicted livestock density and removed biomass

A comparison of MDF-predictions of livestock density against census-based data (Fig. A3) shows that the MDF-predictions mirrored the census-based livestock density data well ($r$=0.68, RMSE=0.45 LU ha$^{-1}$). Both datasets show the highest LU concentrated in the SW England with lower values in the Eastern part of England (Fig. 4). Scotland has consistent areas of high LU in both datasets, in the SW and NE. The GB-average census-based livestock density was 0.76$\pm$47 LU ha$^{-1}$ and respective MDF-predicted livestock density was 0.70$\pm$56 LU ha$^{-1}$. The census-based data (cattle and sheep) for each GB country refer to different years. Livestock census numbers for England, in particular, were recorded in 2010, since when numbers have declined (DEFRA, 2020). This time mismatch with our 2018-2018 estimate could explain the small negative bias (-0.06 LU ha$^{-1}$) between MDF-predicted and census-based LU ha$^{-1}$.

The analysis suggests that the 1855 simulated fields were managed with varying intensity. The majority of simulated fields were grazed-only (75%) and no cut-only fields were simulated. Grazed biomass exceeded cut biomass in 85% of the fields (GCD > 0) and cut biomass exceeded grazed biomass in the remaining 15% (GCD < 0). The mean MDF-predicted annual yield (grazed and cut biomass) was 6$\pm$1.8 tDMha$^{-1}$y$^{-1}$ (5th|25th|75th|95th percentiles : 2.8|4.6|7.3|8.5 tDMha$^{-1}$y$^{-1}$). These results reflect biomass utilisation per grassland management intensity in the GB. Rough grazing grasslands (40% of GB grassland area) have an annual yield (total removed biomass) of 3.09$\pm$1.56 tDMha$^{-1}$y$^{-1}$, permanent grasslands (50% of GB grassland area) have an annual yield of 7.41$\pm$2.02 tDMha$^{-1}$y$^{-1}$ and temporary grasslands (10% of GB grassland area) have a yield of 9.76$\pm$2.03 tDMha$^{-1}$y$^{-1}$ (Qi et al., 2017, 2018).

Most of the MDF-predicted first grass cuts (85%) occurred between the first half of May and the second half of July. For the fields where more than one cut was identified the period between the first and last cut was $\approx$ 2 months. The MDF-predicted day-of-year of first cut increased northwards with the mean date of first cut in northern England and Scotland being 3-6 weeks later than in the southern half of GB. This spatial pattern is likely the combined effect of differences in the onset and duration of the grass growing season and in related management decisions. Due to the small share of cut-and-grazed fields in the simulated

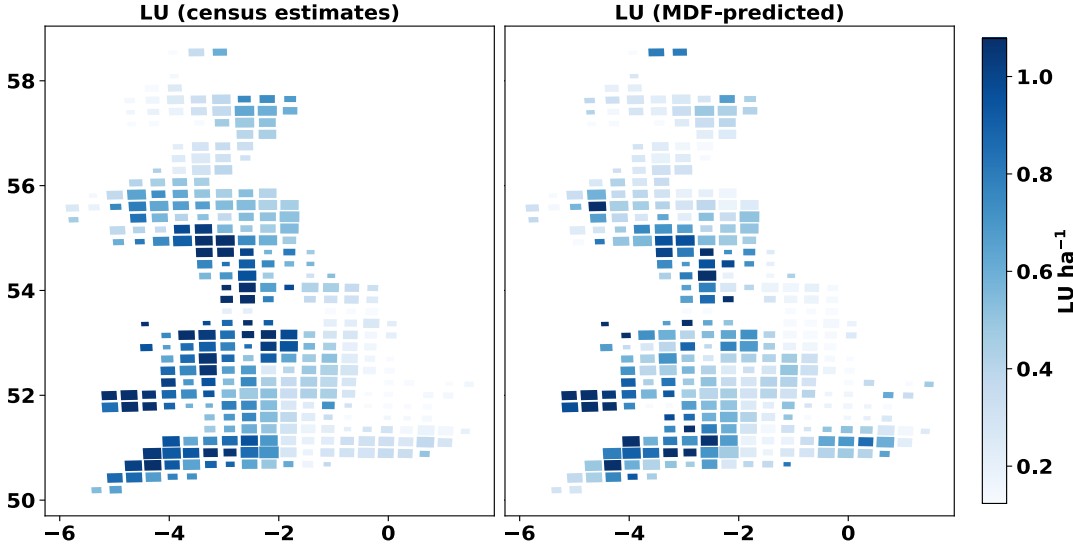

**Figure 4.** Cartograms of census-based and MDF-predicted livestock density (livestock units, LU ha$^{-1}$). The size of cells is adjusted according to the number of simulated fields within it.

dataset, and in order to make the spatial pattern more visible, we present the average month of first cut on a regional basis (Fig.

A4 in Appendix A ).

### 3.3  Predicted C balance and dynamics

MDF-based C cycle estimates show that management affected the C balance of the simulated grassland ecosystems significantly. The difference between grazed and cut biomass volume (GCD) is used to present the impact that these two biomass removal methods have on C balance. The mean annual GPP across GB fields was 30% higher (1992±400 gCm$^{-2}$y$^{-1}$) for

fields where most biomass was removed via grazing (GCD>0) compared to those where cut biomass exceeded grazed biomass (GCD<0) (1518±426 gCm$^{-2}$y$^{-1}$) (Figure 5). Reco was higher for fields dominated by grazing also. The mean NEE across GB was -232±94 gCm$^{-2}$y$^{-1}$, the relative role of grazing compared to cutting did not affect NEE significantly, and 95% of the simulated fields were net C sinks at the ecosystem scale. When considering the role of cutting and grazing C removals and returns to the ecosystem, the impact of cutting as a biomass removal method becomes important. The NBE of fields

dominated by cutting removals (GCD<0) was 38±70 gCm$^{-2}$y$^{-1}$ while fields dominated by grazing removals (GCD>0) had a NBE of -126±95 gCm$^{-2}$y$^{-1}$. On average, 60% more C was removed (grazed and cut) in mostly-cut (GCD>0) grasslands than in mostly-grazed (GCD<0) grasslands. The flux of C into the SOC pool was, on average, 66% larger in mostly-grazed (GCD<0) than mostly-cut (GCD>0) fields. The annual change in the size of the SOC pool ($\Delta_{SOC}$) for mostly-cut (GCD<0)

grasslands was $116\pm52$ gCm$^{-2}$y$^{-1}$ and $36\pm40$ gCm$^{-2}$y$^{-1}$ for grazing-dominated (GCD>0) fields. The spatial distribution of MDF-predicted GPP, Reco, NEE, NBE, removed biomass and C flux into SOC is presented in the cartograms of Figure A5 in Appendix A.

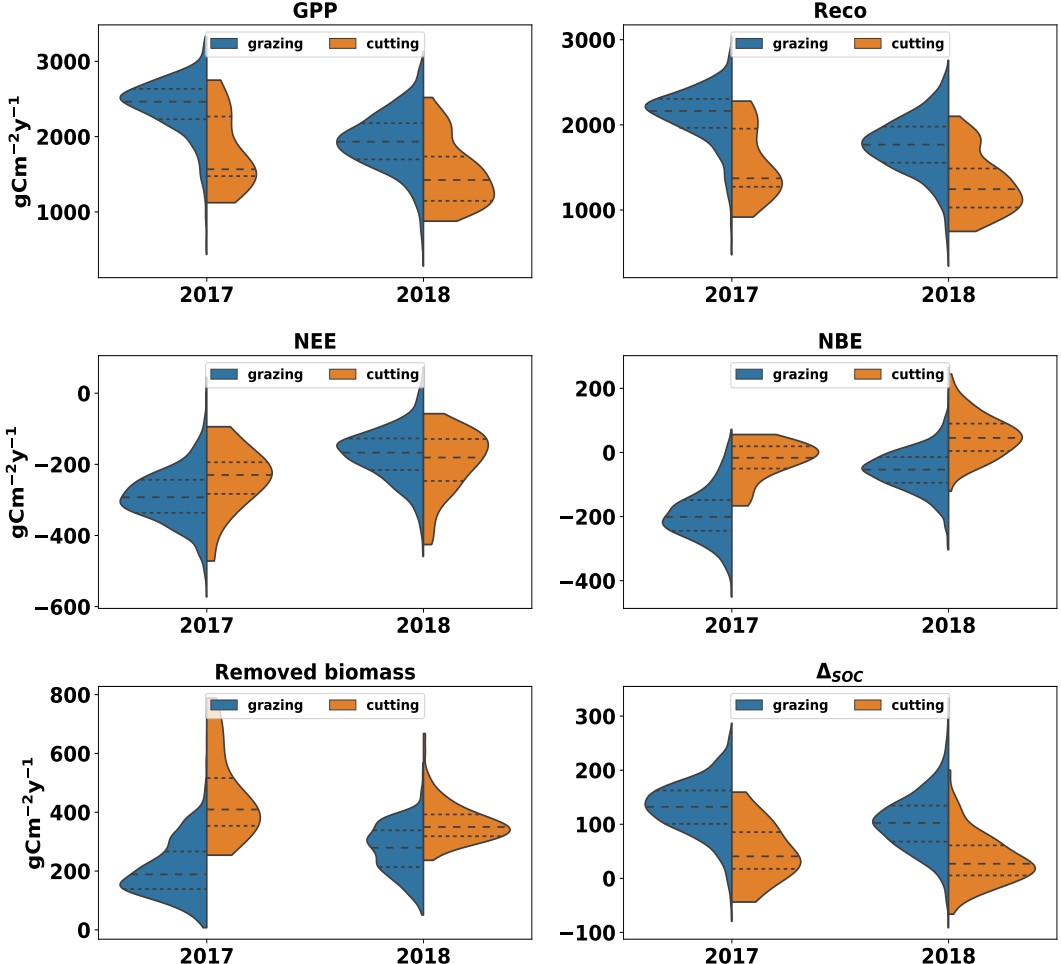

**Figure 5.** Violin plots of GPP, Reco, removed biomass and C flux from litter to SOC based on MDF-predictions (2017-2018) for all simulated fields. Violin plots are split according to whether grazing or cutting removed most grass biomass. The blue side of each violin-plot shows results for fields in which most biomass was removed via grazing (GCD>0). The orange side shows results for fields in which most biomass was removed via cutting (GCD<0).

Seasonal NEE varied across GB, with strongest sinks in Spring and Summer, strongest sources in Autumn, and close to neutral net exchanges during Winter (Fig. A8). However, there were clear inter-annual differences between 2017 and 2018 in the analysis. Across the southern third of GB (the Midlands and Southern England) many grasslands became C sources during the summer of 2018 while remaining areas were weaker sinks than in 2017 A8). This pattern was driven by the 2018 European drought and heat wave, which affected GB as a whole and was particularly acute in the southern half of England (Sibley,

2019). The three-week rolling average VPD in summer 2018 across GB was 50% higher than in summer 2017 (Fig. A7 in supplementary material). The GB-average GPP, $R_a$, $R_h$ and the C flux from the litter to the SOC pool decreased between 2017 and 2018 (Fig. 6). The GB-average flux of C due to litter decomposition in 2018 was less than in 2017 but litter C turnover in

2018 when was nearly two times that of 2018. The GB-average NEE and NBE increased between 2017 and 2018, indicating a reduction in sink strengths (Fig. 5). While only 2% of the simulated grasslands had a NBE>0 in 2017 this share increased 9-fold to 18% in 2018 (Fig. A6 in Appendix A ). The GB-average total removed biomass in the drought-affected 2018 was 27% higher than in 2017. Reductions in cut yields and increases in grazed biomass underlie this increase in GB-average removed biomass (Fig. 5). In this context, the area-mean grazed biomass during the 3 months of spring 2018, in the southern half of GB,

and the 3 months of summer 2018, in the northern half, was higher than the respective seasons in 2017 (results not presented).

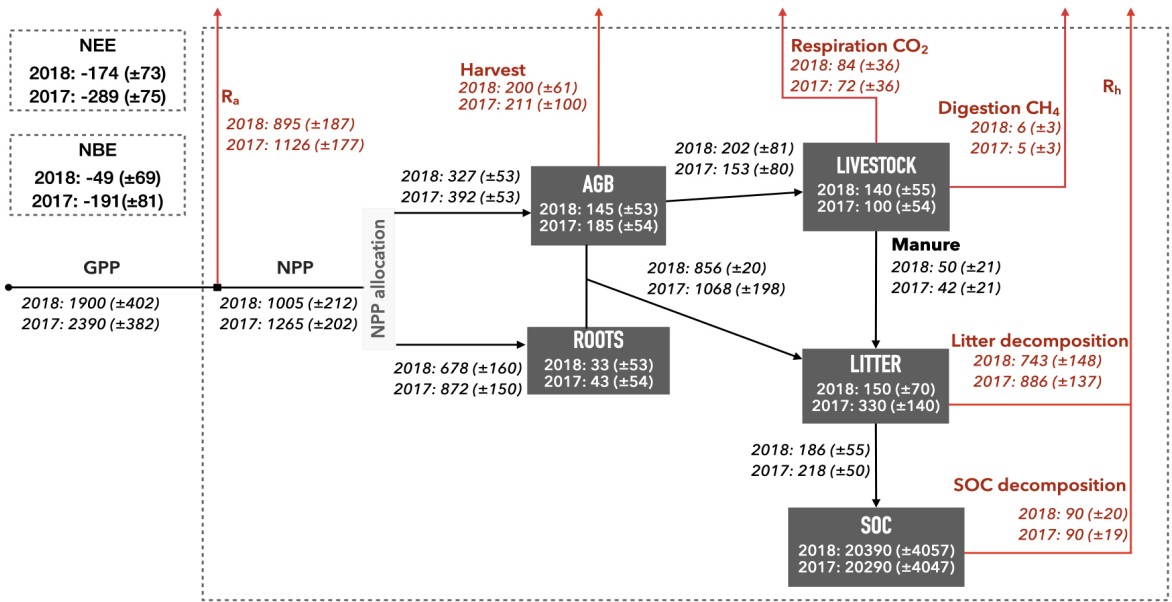

**Figure 6.** Spatial mean ($\pm$ standard deviation) of the estimated C pools and fluxes across the 1855 simulated fields for 2017 and 2018. NEE (net ecosystem exchange) and NBE (net biome exchange) for 2017 and 2018 also presented. Internal ecosystem fluxes are presented as black arrows. Fluxes towards the atmosphere are presented as red arrows. Fluxes values show cumulative g C m$^{-2}$ y$^{-1}$ and pools values show average g C m$^{-2}$ y$^{-1}$. For a schematic of the DALEC-Grass model and all the modelled C pools and fluxes see Figure 1.

## 3.4    Controls on C cycling in GB grasslands

Correlation coefficients (Fig. 8) generated across the 1855 fields show the links between meteorological drivers, key processes (model parameters) and model outputs (C exchanges). There are strong positive correlations between GPP, Reco, $\Delta_{SOC}$ and root:shoot ratio, and these factors are strongly negatively correlated with NBE and NEE. The most productive fields (higher

GPP) are associated with high inputs of C to soils and are the strongest C sinks (more negative NEE and NBE). Among mod-

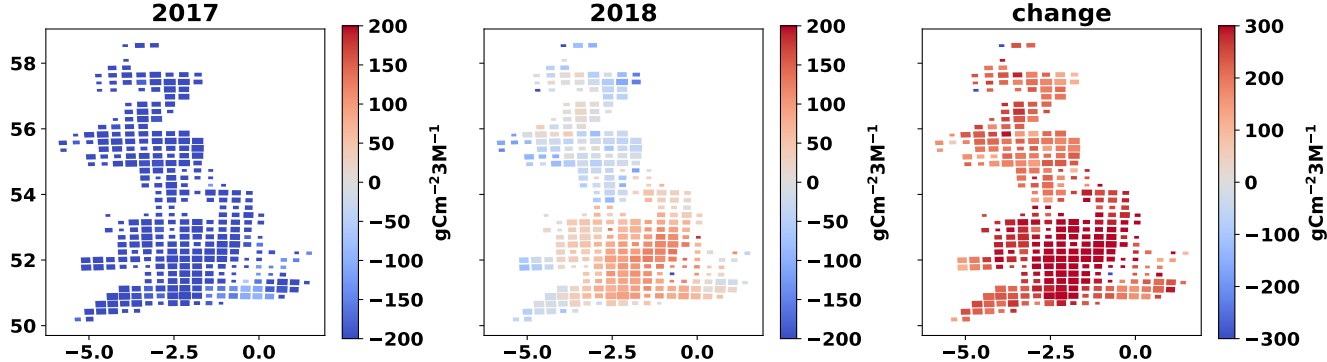

**Figure 7.** Cartograms of cumulative NEE for summer 2017, summer 2018 and change in summer NEE between 2017 and 2018 (cumulative NEE 2018 - cumulative summer NEE 2017). The mean MDF-predicted seasonal NEE of all fields in each cell is presented. The size of cells is adjusted according to the number of simulated fields within it

elled processes, the ratio of C allocation to roots relative to stems and leaves (root:shoot ratio) is the most strongly correlated with the net C balance of the simulated fields (Fig. 8). More-frequent and higher-yielding grass cuts reduce root:shoot ratio and, therefore, reduce the flux of C to litter and, subsequently, to the recalcitrant soil C pool (SOC). The predicted flux of C to the SOC pool has a significant but low positive correlation ($r$=0.38) with the size of SOC pool. Despite that, MDF results show that the volume of C transferred to the SOC pool during the simulated period is, on average, equal to 1 ($\pm$0.25) % of the size of the SOC pool. NBE and NEE are positively correlated to livestock units (LU) and biomass removals, i.e. increases in LU and removals reduce C sinks. Temperature and radiation have relatively weak correlations with NEE and NBE. In contrast, VPD was more strongly related to C fluxes. Higher VPD values correlate with lower GPP and higher NEE and NBE. This positive $r$ for VPD reflects the strong, negative role of the 2018 summer drought.

We expanded on the correlations-based analysis of the MDF results by using (1) the MDF-predicted data on NBE, and (2) the corresponding meteorological drivers and (3) model parameters describing climatic and management controls on grass growth, to train a RF model that estimates NBE. The resulting RF model was able to explain 93% of the variance in MDF-predicted NBE ($r^2$=0.93) using 5 predictors: VPD, mean T, solar radiation, all posterior DALEC-Grass parameters related to climatic effects on grass growth and all posterior parameters related to grassland management. The weight (normalised SHAP) attributed by the RF model to these 5 predictors suggests that management parameters (aggregated weight for all parameters in the group) were the most important factor for grassland NBE over the simulated period (Table 1. The normalised SHAP for management parameters was the highest among the 5 NBE predictors in 2017 (contributed by 34% to NBE) and the second highest (contributed by 38%) in 2018. The 2018 summer heat wave caused the contribution of VPD to NBE to increase from 3% in 2017 to 40% in 2018. Overall, these results reaffirm the conclusions of correlations-based analysis and clarify the importance of grassland management relative to climate and climatic anomalies.

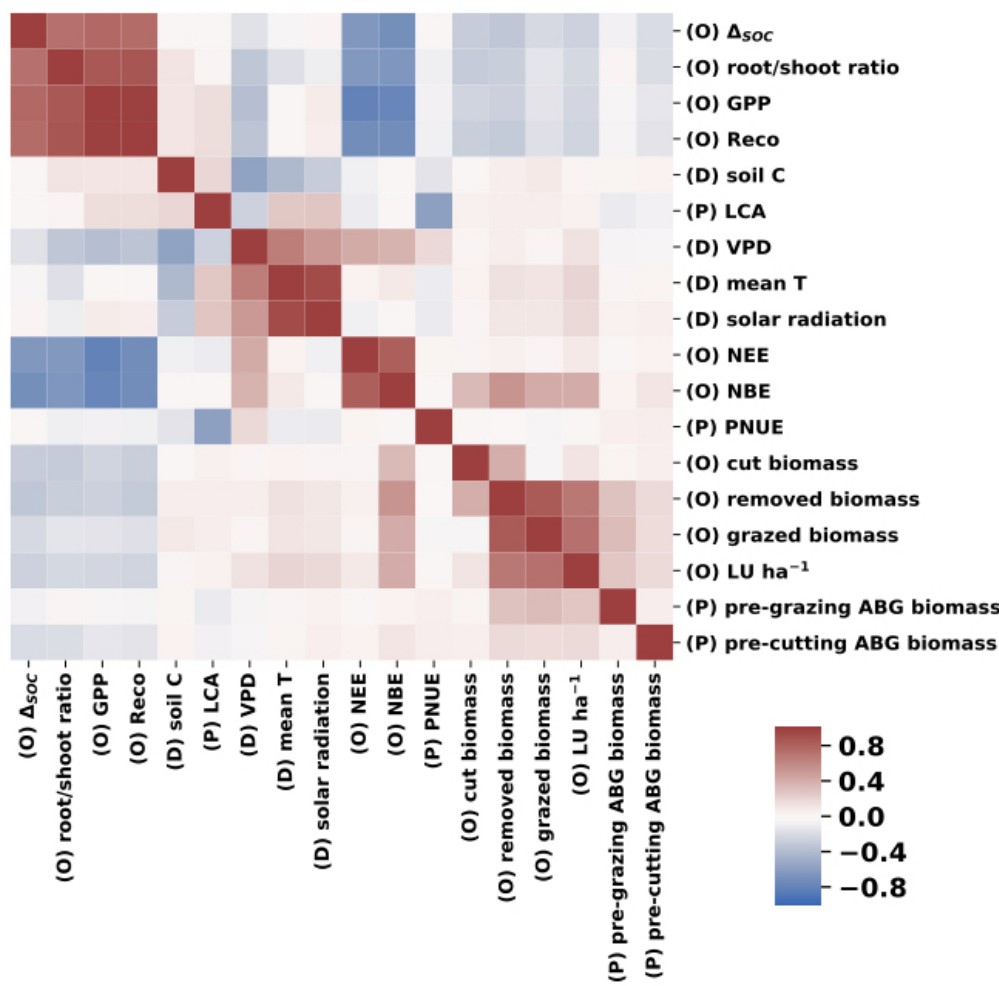

**Figure 8.** Heatmap of correlation coefficients ($r$) between annual mean meteorological model drivers (D), selected model parameters (P) and annual mean MDF predictions (O).

**Table 1.** Normalised SHAP values for RF-based estimation of annual NBE in 2017 and 2018.

| Predictor | 2017 | 2018 |
|---|---|---|
| Climatic effects (parameters) | 0.30 | 0.19 |
| Management effects (parameters) | 0.34 | 0.38 |
| Mean Air Temperature | 0.02 | 0.02 |
| Vapour Pressure Deficit | 0.03 | 0.40 |
| Solar Radiation | 0.31 | 0.01 |

## 3.5 Predictive uncertainty

The size of the uncertainty around MDF estimates is quantified using the RCR (relative confidence range) of MDF outputs. The mean RCR is 42$\pm$9% for LAI, 21$\pm$10% for GPP, 18$\pm$6% for Reco and 26$\pm$16% for grazed biomass (Figure 9). MDF predictive uncertainty is therefore a small faction of the mean estimate of these scalar variables. The GB-average RCR for LAI, GPP and grazed biomass prediction increased from 44%, 26% and 27% in fields where cut biomass did not exceed grazed biomass (GCD>0) to 54%, 40% and 52% in fields where cut biomass exceeded grazed biomass (GCD<0). The higher RCR (mean and SD) for LAI and grazed biomass is caused by the spatio-temporal uncertainty in the vegetation reduction time-series. This input-related uncertainty leads to the MDF algorithm being less effective in identifying cutting instances in some simulated fields; i.e. sampled parameter vectors produce varying predictions on the timing and intensity of grass cuts. The impact of input data uncertainty on RCR is also reflected on the shape of RCR distributions for GPP and grazed biomass (violin plots in Fig. 9). The assimilated EO LAI time-series condition the simulated LAI, which combined with the simulated removals (grazing/cutting), determine the weekly GPP at each simulated field. Reducing the uncertainty (spatial and temporal) around the vegetation reduction data is expected to lead to less variable predictions of cutting timing and intensity and, thus, to lower predictive uncertainty, In general.

## 4 Discussion

### 4.1 Grassland vegetation management across GB

Process modelling combined with earth observation can identify grassland vegetation management effectively over large spatial domains. The distribution of MDF-based livestock densities across GB mirrors the independent determined census-based numbers of cattle and sheep per area; MDF estimated livestock density = 0.7 ($\pm$0.56) LU ha$^{-1}$, census based livestock density = 0.76 ($\pm$0.46) LU ha$^{-1}$. Considering that grasslands with a corresponding LU ha$^{-1}$ < $\approx$ 0.5 are thought as supporting a low livestock density and those with LU ha$^{-1}$ > $\approx$ 1 as supporting a high livestock density, the average managed grassland in GB supported an intermediate livestock density in 2017-2018 (Chang et al., 2015b).

The MDF-predicted GB-average pasture dry matter yield (6$\pm$1.8 tDMha$^{-1}$y$^{-1}$) is within the range for GB permanent pastures (7.41$\pm$2.02 tDMha$^{-1}$y$^{-1}$) as estimated by Qi et al. 2017 using statistical extrapolation of field-measured data. Due to the field-size limits (6-13ha) used in sampling for fields across GB, the share of less intensively managed grasslands is likely biased high in the simulated fields' dataset (Qi et al., 2018). Cut-only grasslands were under-represented in our analysis, as we expected $\approx$ 10% of field to be in temporary management for cutting. We believe this is an artefact of the noise in the vegetation reduction time-series, which led to cut-only grasslands failing to pass the 50% overlap limit and being excluded from the final dataset. The inclusion of fields that are 6-13ha in size could have led to an under-representation of cut-only fields but we note that no data exist on the percentage of managed grasslands that are grazed-only, cut-and-grazed and cut-only.

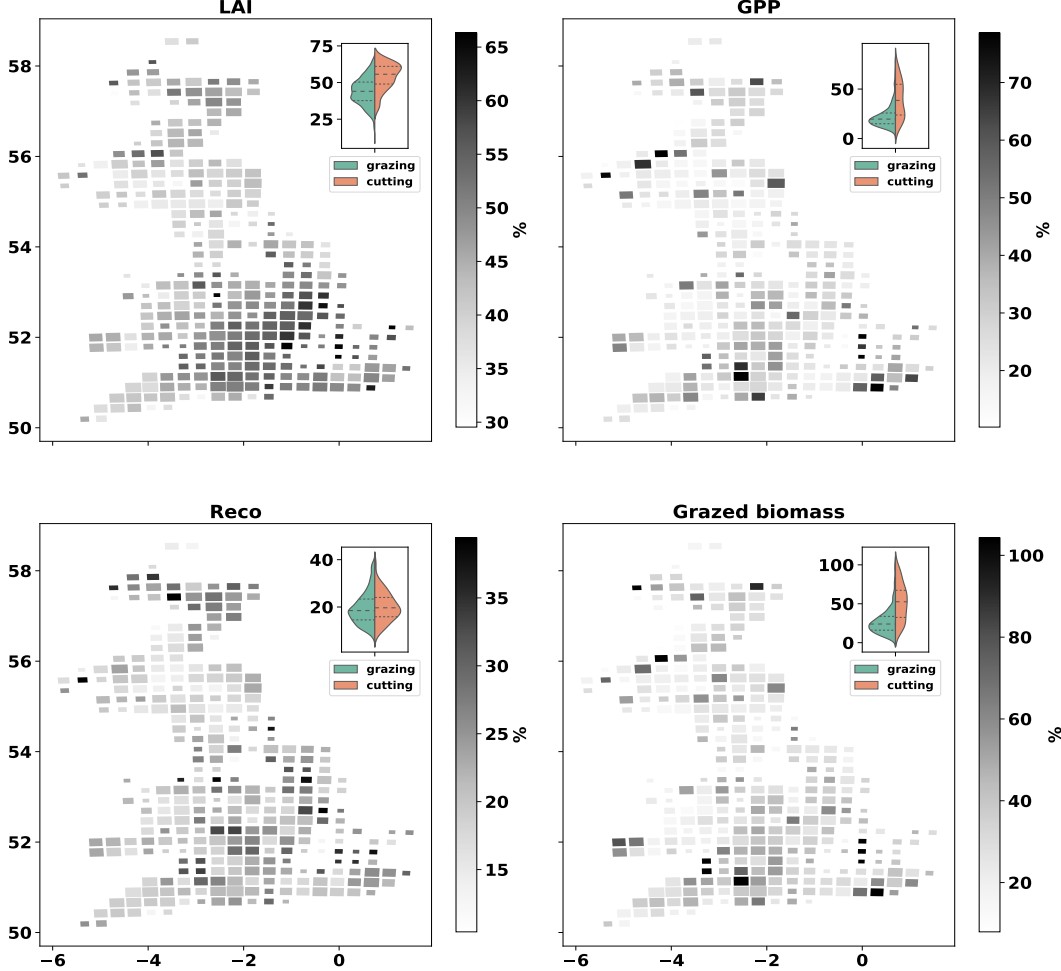

**Figure 9.** Cartograms of relative confidence range (RCR, $100 \times CI \div$ mean) of MDF-predicted LAI, GPP, Reco and grazed biomass. The mean across all fields in each cell is presented. The size of cells is adjusted according to the number of simulated fields within it. The violin-plot insets present the distribution of MDF estimates grouped according to the relative contribution of grazing and cutting to the total annual biomass removal (GCD). The green side of each violin-plot shows results for fields in which most biomass was removed via grazing (GCD>0). The orange side shows results for fields in which most biomass was removed via cutting (GCD<0).

## 4.2 The C balance of managed grasslands

The presented results are probabilistic model-based estimates produced by carefully upscaling our quantitative understanding of grassland C cycling under GB conditions. DALEC-Grass is a process-based C biogeochemical model that has been calibrated and validated against in situ data on C pools and fluxes collected over 11 years and at two variably-managed grassland sites in GB (Myrgiotis et al., 2020). This measured dataset is to our knowledge the most detailed and extensive available for managed grasslands in GB. It includes measured data on NEE, soil C, soil surface respiration, above and below-ground biomass-C, harvest yields, LAI and human management-related factors (i.e. livestock details, manure and fertiliser use). The credibility of DALEC-Grass estimates of C fluxes is established on previously published work on model validation and testing of the grazing and cutting inference algorithm (Myrgiotis et al., 2021).

The results of this study show that the majority of managed grasslands in GB were net C sinks during 2017 (NEE=-289±76 $gCm^{-2}y^{-1}$) and 2018 (NEE=-174±74 $gCm^{-2}y^{-1}$) at the ecosystem level, i.e. based on $CO_2$ gas exchanges. Numerous flux tower-based studies have concluded that managed temperate grasslands are, on average, C sinks but NEE estimates vary greatly between $\approx$ -700 $gCm^{-2}y^{-1}$ to almost C-neutrality (Soussana et al., 2007; Gilmanov et al., 2007; Skinner, 2008). The scale of NEE increase between 2017 and 2018 is comparable to past field-based estimates under normal and heat-wave conditions (Klumpp et al., 2011). When considering C fluxes that also included grazing/cutting removals and manure return to the soil (i.e. NBE), the simulated grasslands were net C sinks during 2017 (NBE=-191±82 $gCm^{-2}y^{-1}$) and close to C neutral in 2018 (NBE=-49±70 $gCm^{-2}y^{-1}$). These NBE estimates are comparable to those in the literature but the inconsistency in the variables included in NBE calculations makes comparisons less than straightforward (Soussana et al., 2007; Skinner, 2008). Based on RF-based analyses of climate drivers and model parameters we argue that the increase in mean annual NEE and NBE between 2017 and 2018 was caused by the 2018 summer heatwave. The negative effect of elevated annual temperatures on grassland biomass productivity and NEE has been examined in measurements and model-based studies on European grasslands before (Jansen-Willems et al., 2016; Ciais et al., 2005; Jansen-Willems et al., 2016; Thompson et al., 2020). The mechanistic understanding, and the model representation, of how plants respond to increased VPD is improving but key aspects are still disputed (Grossiord et al., 2020; Massmann et al., 2019).

Our study shows that biomass removals were key determinants of the C balance of managed grasslands. The role of cutting relative to grazing as a biomass removal method was found to be particularly important. Grasslands in which most biomass was removed via cutting had a lower GPP and Reco as opposed to grasslands in which grazing was the main biomass removal method (Fig. 5). However, when GPP and Reco are summed the resulting NEE did not vary significantly between mostly grazed and mostly cut grasslands. All of the simulated grasslands were grazed and underwent more or less frequent defoliation during the simulated period. When cutting occurs the leaf area of a grassland is reduced close to zero, which represents a diminution of the grassland's photosynthetic capacity. According to DALEC-Grass, in the post-cutting period the simulated grassland allocates almost all of its C to aboveground tissues (stems, leaves) in order to build up the leaf area necessary to increase photosynthetic activity and sustain growth. This causes a smaller root-to-shoot ratio in grazed and cut grasslands compare to grazed-only ones as well as a reduction of root-based C inputs to the litter pool (lower $R_h$). Grazing that occurs during the

475 post-cut period maintains the volume of leaves at relatively low levels. This leads to a reduced annual GPP for grasslands that are both grazed and cut and also means lower manure-C returns to the soil, which explains the weaker C sinks (higher NBE) of most grazed-and-cut grasslands compared to grazed-only fields. Taking into account the fact that most GB grasslands undergo alternations between cutting and grazing and defoliation patterns vary from year to year highlights the importance of detecting these patterns (Fetzel et al., 2017).

## 480    4.3   Factors that control managed grassland C dynamics

Management and climate have a combined effect on C dynamics and disentangling their individual impacts is a challenge (Ammann et al., 2020). Here, we used a correlation matrix of model drivers, parameters and outputs to understand how climate and management affect the predicted C fluxes and balance of the simulated grasslands. The correlation matrix revealed a negative effect (r<0) of VPD on GPP and Reco, and a positive effect (r>0) on NEE and NBE. Biomass removals had a similar
effect with higher removals corresponding to lower GPP and Reco and more positive NEE and NBE. Moreover, we used the model's climate drivers and all of its management and climate-related parameters to train a RF model that estimates NBE. The analysis of the RF model structure showed that the contribution of VPD as a RF predictor of NBE increased from 3% in 2017 to 40% in 2018, with VPD being the most important determinant of NBE in 2018. This increase is linked to the heat wave and drought conditions during that year. Management-related parameters were the most important determinant of NBE in 2017 and
the second most important in 2018.

The conclusions that we draw in regards to which factors have more influence on grassland C dynamics are based on two assumptions. Firstly, we assume that the simulated grasslands are well-optimised for the intended use; i.e. to sustain different types of livestock (e.g. dairy and/or beef cattle and/or sheep). This means that each sward is maintained in good condition and that farmers manage their fields optimally based on their long-term experience. Secondly, the fact that a large share of the
495 simulated fields (especially in the southern half of England) experienced continuous weeks of unusually hot and dry weather conditions during one (2018) of the two simulated years is treated as a climate anomaly; i.e. climate in 2018 is not representative of normal climatic controls on C balance. Based on these assumptions, we argue that the simulated vegetation management, as inferred from the observational data, was adapted to the seasonal weather anomaly. Therefore, significant changes in ecosystem C cycling were beyond the control of human management and can be mostly attributed to the seasonal weather anomaly.

Our findings on the role of management are in agreement with findings in a number of relevant studies, notwithstanding differences in methodologies and eco-climatic conditions. Skinner 2008 found that higher biomass removals increase NBE based on C flux measurements in cut-and-grazed temperate grasslands in the USA. Koncz et al. 2017 used eddy covariance measurements of C fluxes at a cut-only and a grazed-only field in Hungary and found that the cut field had a more positive NBE (smaller sink) than the grazed field. Senapati et al. 2014 reached the same conclusion as Koncz et al. 2017 using eddy
covariance measurements from a cut-only versus grazed-only experiment in France. Soussana et al. 2010 reviewed studies on European managed grassland C balance and found that grazed-only grasslands had the lowest NBE, followed by cut-only grasslands with cut-and-grazed grasslands having the highest NBE (NBE in this study included animal methane-C and C-leaching fluxes). Based on eddy covariance measurements over two years at three grassland sites with varying management

intensity in Switzerland Zeeman et al. 2010 concluded that management (including biomass removals and manure application) has a strong influence on C fluxes and balance.

In summary, we conclude that management is a key determinant of the C balance of managed grasslands in GB. We note that climatic anomalies, such as heat waves and droughts, can reduce the relative importance of management as a determinant of grassland C balance. In simple terms, human decisions can adjust grassland sink or source strength and this depends mostly on the soil's existing C stock, the sward's composition and condition and the timing and intensity of livestock grazing and grass cutting. Climate change can change this fine C balance substantially and prolonged heat and drought is one way in which this can occur in regions with temperate maritime climate.

### 4.4 Predictive uncertainty

We use the RCR to quantify the uncertainty around the MDF-predicted variables. RCR shows how wide the 95% confidence intervals (i.e. $2\times$SD, assuming normality) are relative to the mean value. The assimilated LAI data come from processing S2 images and have an uncertainty attached to them. This observational uncertainty is not always examined in relevant studies but a relative SD of 15% is considered as representative (Zhao et al., 2020). This means that the average RCR of the assimilated observational LAI data is 30%. Considering that MDF predictions incorporate model parametric uncertainty as well, the mean analysis LAI RCR of 45% is, as expected, larger than, but of similar magnitude to, the observational uncertainty of 30% (Fig. 9).

The estimated predictive uncertainty for LAI, GPP and grazed biomass was noticeably higher for fields that were mostly cut (GCD<0) (Fig. 9). The MDF algorithm does not infer cutting simply by translating large reductions in vegetation as cuts. The MDF algorithm examines each weekly vegetation reduction input to decide whether to simulate it as cutting, grazing or ignore it depending on the simulated amount of foliar biomass at the time. The simulated amount of foliar biomass is constrained through the assimilation of field-specific EO-based LAI time-series. A weekly vegetation reduction will be simulated as a cut when it is reasonable both biophysically and agronomically based on the EDCs (see section 2.1.2). This higher predictive uncertainty when cutting occurs suggests that the best way to obtain more accurate predictions is to improve the spatial and temporal resolution of the vegetation reduction time-series and/or estimates of LAI. Using radar (e.g. Sentinel-1) to derive LAI in spite of cloud cover would be a valuable advance.

### 4.5 Limitations

This study uses a MDF algorithm that depends on EO data and process modelling of C dynamics in grasslands. The Proba-V-based vegetation reduction time-series that are used to drive DALEC-Grass have a resolution (9ha) that is coarse when compared to the average size of grassland fields in GB. These noisy data on vegetation reduction cause increased uncertainty in MDF predictions especially in regards to the timing of cutting events. Moreover, most areas of GB are affected by frequent cloudiness, which means that the number of Sentinel 2-based LAI data points per year and simulated field is limited compared to other parts of the world. However, we ensured 30 images per field over two years in our selection process, and this richness of information at field resolution and for national domains is unprecedented in such an analysis.

DALEC-Grass was developed and tested under GB conditions, showing high skill in predicting C allocation and $CO_2$ fluxes under variable management and different soil conditions (Myrgiotis et al., 2020). However, DALEC-Grass can only infer effects on grass growth through the processes it simulates, and so can mis-attribute effects arising from missing processes. For instance, the MDF algorithm can adjust a specific plant growth rate parameter (DALEC parameter P10, photosynthetic N use efficiency) between fields based on observed LAI dynamics and weather. Inferred P10 variation among fields might be linked to spatial patterns in soil fertility. But because there is no direct soil moisture constraint on LAI in DALEC-grass to be adjusted, a real spatial soil moisture limitation on LAI might be mis-interpreted as a restriction on P10. So we should be cautious in interpreting process variation and assigning with certainty to a particular forcing. Also, a single P10 estimate is made for each field covering both 2017 and 2018, so the current analysis does not allow field nutrient supply (and therefore P10) to change between years. The strong differences in sink strengths observed in the 2017 and 2018 analyses are informative. The flux differences cannot arise from parameter differences between years, as these parameters are constant. Instead differences must arise from process changes (e.g. GPP) resulting from changes to the forcing (VPD, Fig. 1) and changes to the assimilated data (LAI) between years. The larger LAI uncertainty in southern GB (Fig. 9) may be related to soil moisture impacts on grass growth that we fail to identify with the current model structure. Finally, DALEC-Grass has been validated against data from grasslands dominated (>90%) by perennial ryegrass (*Lolium perenne*) and its ability to simulate swards with larger shares of herbs and forbs has not been tested.

In general, the ability to use field-specific observed information on key aspects of grassland vegetation and to infer vegetation grazing and cutting are the key advances presented in this study. The majority of grassland-focused model-based estimates for large domains typically rely on uncertain information on grazing and cutting. Also, with few recent exceptions, most relevant studies do not include field-specific validation of model predictions, which results in highly uncertain estimates. On the other hand, the calculation of some lateral flows of C (manure input) remains an outstanding challenge as these depends on information that cannot be inferred from EO-based time-series. The size, type and age of livestock significantly affect aboveground biomass and the turnover of grazed biomass-C (Bahn et al., 2008). While we cannot infer such detailed livestock information our probabilistic MDF framework allows us to attribute uncertainty to livestock C turnover and, thus, to quantify their impact on predictions. This attribution can be done by treating grazed biomass-C conversion factors (Fig. 1) as model parameters.

Our approach cannot detect human-managed manure application from EO data and so does not consider this method of manure-C addition to the soil. Grazed biomass-to-manure C conversion factors are used in DALEC-Grass to estimate the amount of manure-C produced and added to the simulated soil litter pool at each time step. This way of applying manure is a simplification of what happens in reality where grazing livestock will deposit some manure to the soil while some of their manure will be collected during periods of housing and stored to be applied across a farm's fields and/or traded to other grassland or arable farms. Typically, most manure is applied in GB grasslands during Spring and Autumn. Despite all this, GB livestock is primarily grass fed and the volume of manure produced in a farm is directly related to the biomass productivity and the livestock density maintained at the different farm fields, which MDF can detect and deduce respectively (Smith and Williams, 2016). We almost certainly mis-estimate the volume of manure-C returned to soils annually (2017 and 2018) at the

1855 fields that were simulated due to lateral transfers. However, the predicted spatial distribution of manure production and application intensity is likely to be representative of reality since this should be following the spatial distribution of livestock density and biomass productivity ; factors that MDF predicted well (section see 3.2).

### 4.6 Future work

Our overarching aim is to produce a computational ecosystem modelling framework that is (1) able to utilise the swathes of EO data that are increasingly becoming available while (2) being easy to adapt and incorporate new knowledge gained from field/lab experiments and observations. This study showed that the MDF algorithm will benefit most from improving the temporal resolution and quality of EO LAI data used. We believe that by advancing on this front the algorithm will be able to produce more accurate estimates across grasslands in Europe and other regions with similar agro-climatic conditions. Introducing soil moisture and N cycling-related processes to DALEC-Grass will pave the way for more detailed consideration of the effects of fertiliser use and different grass mixtures, and for its application at climatically-critical rangelands and pastures across the world (e.g. tropical and dry regions). DALEC-Grass has a structure that facilitates the incorporation of modelling advances made with other DALEC-based models such those presented in Revill et al. 2021 for foliar N and Smallman and Williams 2019 for soil moisture. We note that the quality of soil C-related data is critical for better constraining below-ground C pools and fluxes (e.g. heterotrophic respiration) and so is the availability of ground measured data on field-scale C fluxes (e.g. NEE). Improvements in the quality, volume and availability of relevant spatial data in the future will improve the credibility for MDF estimates.

### 5 Conclusions

This study presented how, by fusing EO data and biogeochemical modelling of managed grassland C dynamics at field resolution across a national domain, a MDF framework can detect biomass removals and use this information to predict grassland C fluxes and balance probabilistically. In addition, the study showed how field-specific model predictions of grassland vegetation can be validated against field-specific EO-based LAI time-series. We argue that both of these uses of EO data in model-based studies represent key advancements that increase the credibility of field-scale estimates of C dynamics in managed grasslands. Our results show that MDF-predicted annual yields and livestock density mirror ground based information well. In agreement with a range of studies on temperate grasslands in Europe and beyond, our study reaffirms the C sink potential of managed grasslands in GB. In contrast to previous measurements and model-based studies, however, we showed how MDF can quantify and interpret C dynamics across a large domain (GB) while also resolving sub-field scale variability in vegetation management. This granularity is vital as our results show how management differences between fields have strong effects on net C balance. It is widely accepted that climate change is manifesting itself, among other ways, as more frequent droughts in northern Europe (Peters et al., 2020). Our study showed how the most prolonged drought (2018) that has been recorded in GB since 2000 affected the C balance of managed grasslands. It highlights that the ability of temperate maritime grasslands to sequester C could be significantly affected by prolonged heat waves and drought. Various climatic and management-related factors affect

both the annual C balance and the seasonal grassland biomass utilisation in livestock farming in GB; and northwest Europe in general.

National targets for C neutrality in the agricultural sector and the unfolding of climate change create a challenging future for GB grassland farming. The estimation of grassland C balance using MDF has a number of limitations, including the lack of field-scale data on soil C and fertiliser and manure application across large domains. Yet, these limitations can be addressed if MDF is used as part of a land C management monitoring system, in which farmers report field-scale activity data (i.e. fertiliser and manure use) and measure soil C for validation purposes. Overall, the strength of probabilistic MDF is its potential to utilise disparate observational data and provide estimates with well defined uncertainties. In this respect, the volume and resolution of observational data on plant and soil conditions of grasslands continuous to grows driven by advances in EO science and infrastructure and by an increasing interest and investment in environmental monitoring technologies (e.g. low cost proximal sensing, integrated network of sensors and stations). We argue that in the near future farmers and governments alike will be able to benefit from MDF approaches that provide key monitoring tools for C balance, and guidance on adaptation and mitigation of climate change effects on agriculture towards meeting net-zero goals.

*Author contributions.* VM and MW devised the study concept. VM developed DALEC-Grass, implemented the MDF and undertook the analysis with support from all authors. VM led the writing, with support from MW and LS.

*Competing interests.* The authors declare no competing interests

*Acknowledgements.* This study was supported by the Natural Environment Research Council (NERC) of the UK through several projects: the Soils Research to deliver Greenhouse Gas REmovals and Abatement Technologies (Soils-R-GGREAT) project (NE/P018920/1), DARE-UK (NE/S003819/1), and GREENHOUSE (NE/K002619/1). MW acknowledges support from NCEO and the Royal Society. We acknowledge the inputs and support of the CARDAMOM development team who contributed to the algorithm concept. We thank Anthony Bloom (NASA Jet Propulsion Laboratory) for his support.

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

**Table A1.** DALEC-Grass parameters (number, description, units and prior min/max values)

| Code | Description | Unit | min | max |
|---|---|---|---|---|
| P1 | Decomposition rate | fraction d$^{-1}$ | 0.0011 | 0.0136 |
| P2 | Fraction of GPP that is respired | - | 0.46 | 0.48 |
| P3 | Growing Season Index (GSI) sensitivity for leaf growth | - | 0.75 | 0.90 |
| P4 | NPP belowground allocation parameter | - | 0.55 | 0.70 |
| P5 | Maximum GSI for leaf turnover | - | 1.34 | 1.99 |
| P6 | Turnover rate of roots | fraction d$^{-1}$ | 0.052 | 0.071 |
| P7 | Turnover rate of litter | fraction d$^{-1}$ | 0.022 | 0.049 |
| P8 | Turnover rate of soil organic matter | fraction d$^{-1}$ | 4E-07 | 1.26E-05 |
| P9 | Temperature Q10 factor | - | 0.047 | 0.067 |
| *P10 | Photosynthetic N use efficiency (PNUE) | g C per g N per leaf m$^2$ per day | 21 | 25 |
| P11 | Maximum GSI for labile/stem turnover | - | 0.40 | 0.67 |
| P12 | Minimum GSI temperature threshold | K | 230 | 243 |
| P13 | Maximum GSI temperature threshold | K | 279 | 296 |
| P14 | Minimum GSI photoperiod threshold | seconds | 9580 | 15590 |
| *P15 | Leaf Mass C Area | g C per m$^2$ of leaf | 45 | 52 |
| P16 | Initial C in stem/labile pool | g C m$^{-2}$ | 20 | 35 |
| P17 | Initial C in foliar pool | g C m$^{-2}$ | 85 | 100 |
| P18 | Initial C in roots pool | g C m$^{-2}$ | 40 | 355 |
| P19 | Initial C in litter pool | g C m$^{-2}$ | 250 | 790 |
| P20 | Maximum GSI photoperiod threshold | seconds | 33200 | 40000 |
| P21 | Minimum GSI vapour pressure deficit threshold | Pa | 100 | 350 |
| P22 | Maximum GSI vapour pressure deficit threshold | Pa | 1000 | 1500 |
| P23 | Critical GPP for LAI increase | g C m$^{-2}$ d$^{-1}$ | 0.035 | 0.153 |
| P24 | GSI sensitivity for leaf senescence | - | 0.993 | 0.996 |
| P25 | GSI growing stage indicator | - | 0.72 | 1.01 |
| P26 | Initial GSI value | - | 1.56 | 1.73 |
| *P27 | Pre-grazing AGB threshold | g C m$^{-2}$ | 50 | 100 |
| *P28 | Pre-cutting AGB threshold | g C m$^{-2}$ | 120 | 160 |
| P29 | Leaf to stem allocation parameter | - | 0.6 | 0.7 |
| P30 | Post grazing labile/stem loss | - | 0.01 | 0.03 |
| P31 | Post cutting labile/stem loss | - | 0.50 | 0.53 |

* parameters for which the prior was wider than suggested by Myrgiotis et al. 2021.

**Table A2.** Ecological and Dynamic Constraints (EDC)

| No | Explanation | Reference |
|---|---|---|
| 1 | The turnover rate of the soil organic matter pool cannot be faster than that of the litter pool | |
| 2 | Initial SOC pool cannot be < the sum of all other pools (litter, roots, aboveground) | |
| 3 | The soil organic matter pool cannot lose or gain > 5% of its C in a simulated year | |
| 4 | Annual GPP and ecosystem respiration cannot be <800 g C m$^{-2}$ or >2800 g C m$^{-2}$ | (Xia et al., 2015; Gilmanov et al., 2007) |
| 5 | Weekly mean GPP cannot be >25 g C m$^{-2}$ | (Xia et al., 2015; Gilmanov et al., 2007) |
| 6 | Cutting yield cannot be <80 g C m$^{-2}$ or >385 g C m$^{-2}$ | (Qi et al., 2017) |
| 7 | No more than 4 cuts can occur each simulated year | (Qi et al., 2017) |

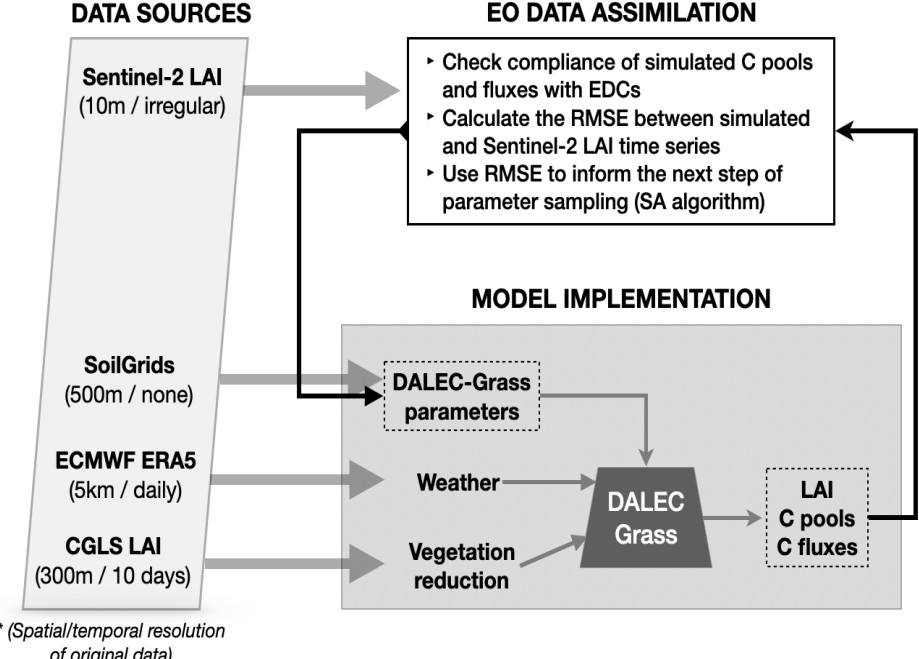

**Figure A1.** Schematic description of data sources and data flow in the model-data fusion process. The DALEC-Grass model is driven by weekly weather and vegetation reduction data (see sections 2.1.5 and 2.1.4 ). The initial size of the soil organic C (SOC) pool for each simulated field is obtained from the SoilGrids database (Hengl et al., 2017) and used as a DALEC-Grass parameter (with uncertainty attributed to it). DALEC-Grass produces outputs on weekly C pools, fluxes and removals (see Fig. 1). It also produces weekly time-series of LAI. Observational time-series on LAI are assimilated by reducing the RMSE between observational and simulated LAI time-series (see section 2.1.4). The assimilation is performed in CARDAMOM by using the simulated annealing (SA) method/algorithm (see section 2.1.3).

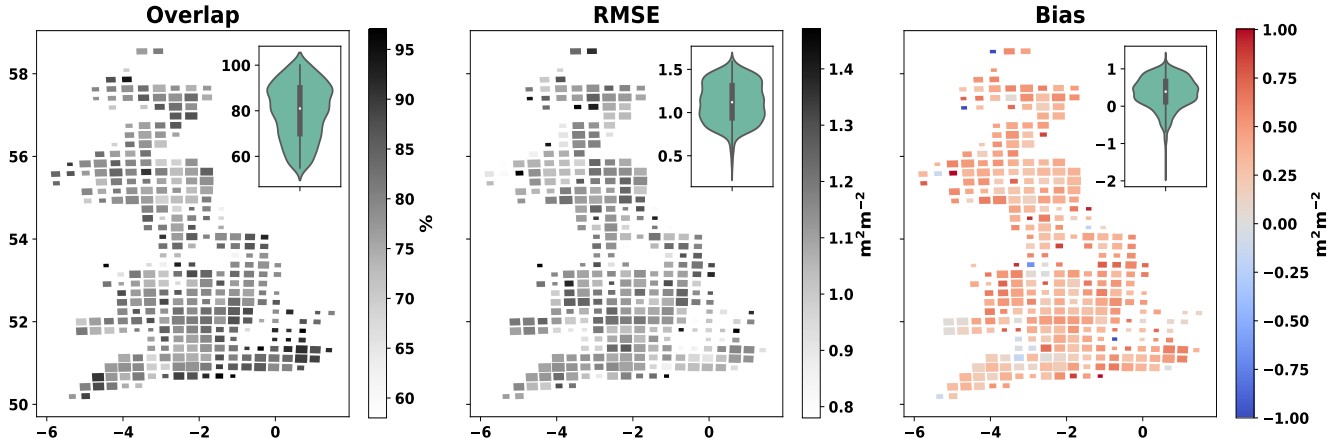

**Figure A2.** Cartograms of overlap (%), root mean square error (RMSE) ($m^2m^{-2}$) and bias ($m^2m^{-2}$) between MDF-predicted and assimilated LAI (EO-based). The size of cells is adjusted according to the number of simulated fields within it. The violin-plot insets present the distribution of each evaluation metric across all simulated fields.

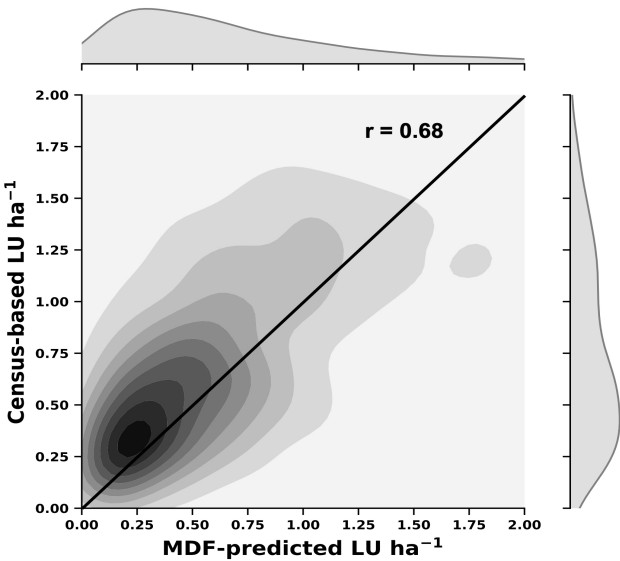

**Figure A3.** Kernel density estimates plot (inner part) and distributions (outer part) of MDF-predicted (x-axes) and census-based (y-axes) livestock density.

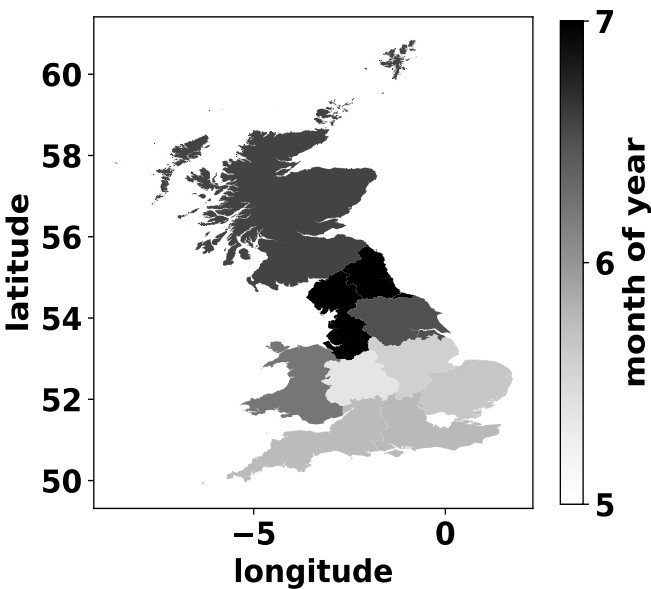

**Figure A4.** Mean month-of-year of first simulated grass cutting per GB region.

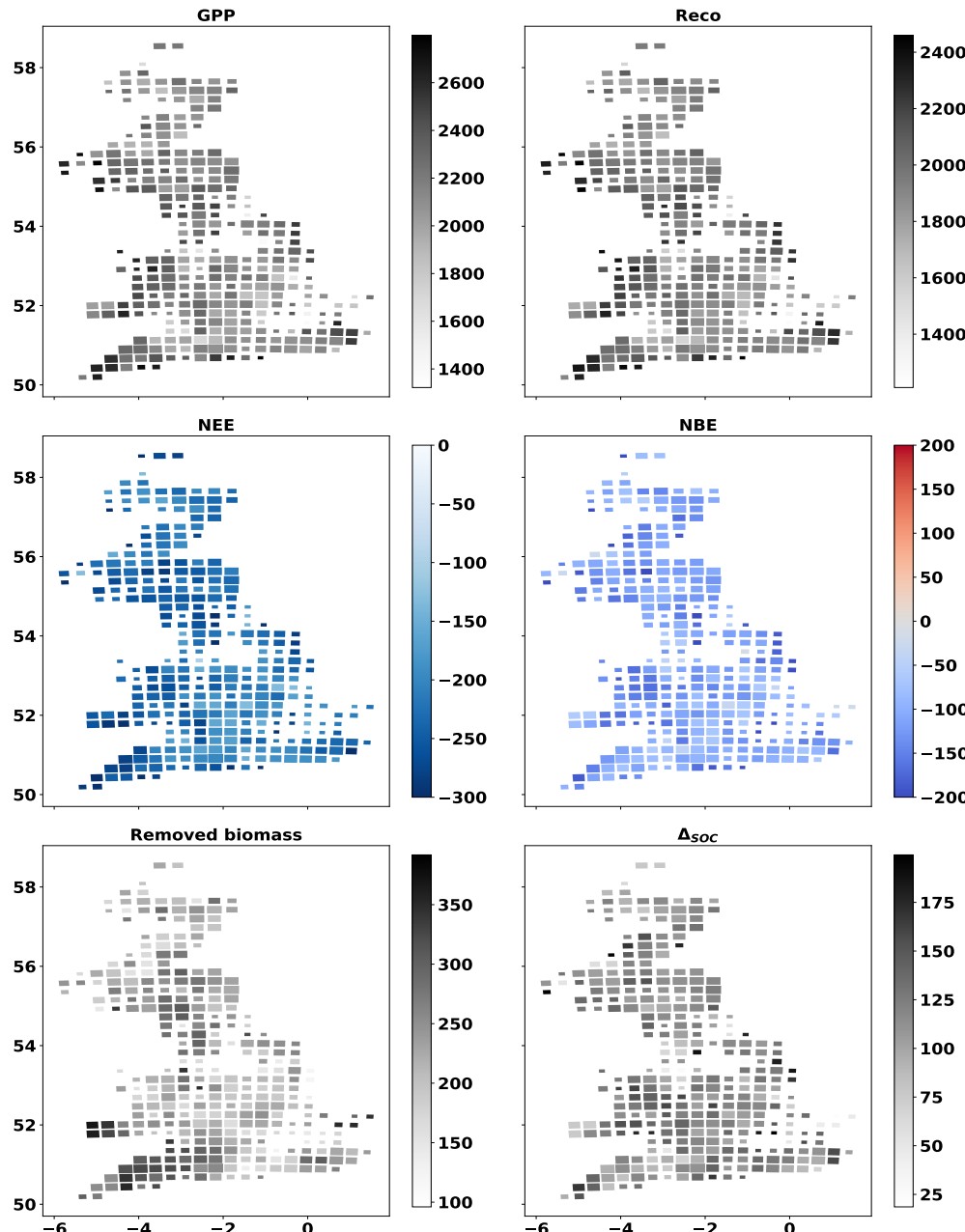

**Figure A5.** Cartograms of MDF-predicted GPP, Reco, NEE, NBE, removed biomass and C flux to SOC. The mean value for 2017-2018 across all fields in each cell is presented. The size of cells is adjusted according to the number of simulated fields within it. Unit : $gCm^{-2}y^{-1}$

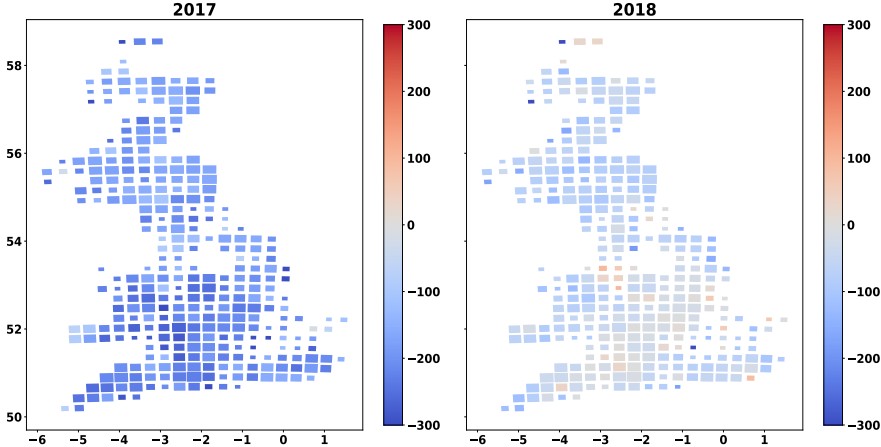

**Figure A6.** Cartograms of MDF-predicted NBE for 2017 and 2018. The mean across all fields in each cell is presented. The size of cells is adjusted according to the number of simulated fields within it. Unit : $gCm^{-2}y^{-1}$

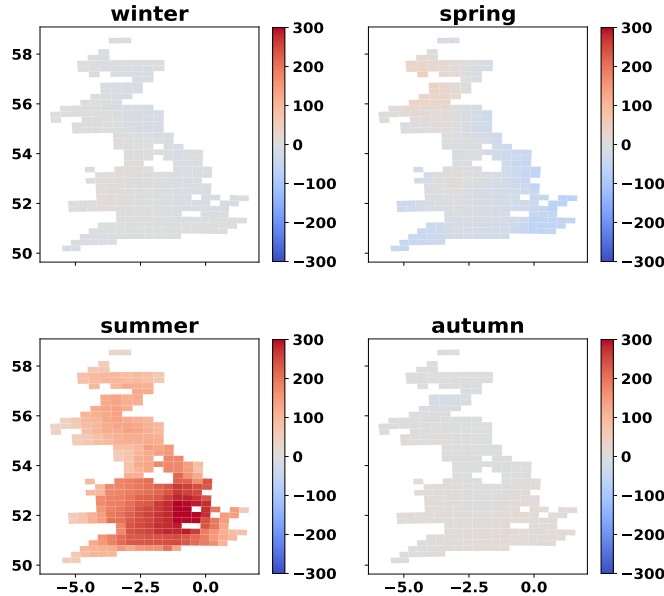

**Figure A7.** Map of inter-annual (2017-2018) difference in three-week average VPD (Pa) per season. The map is a 25km grid of GB. Only grid cells that contain at least one simulated field are presented.

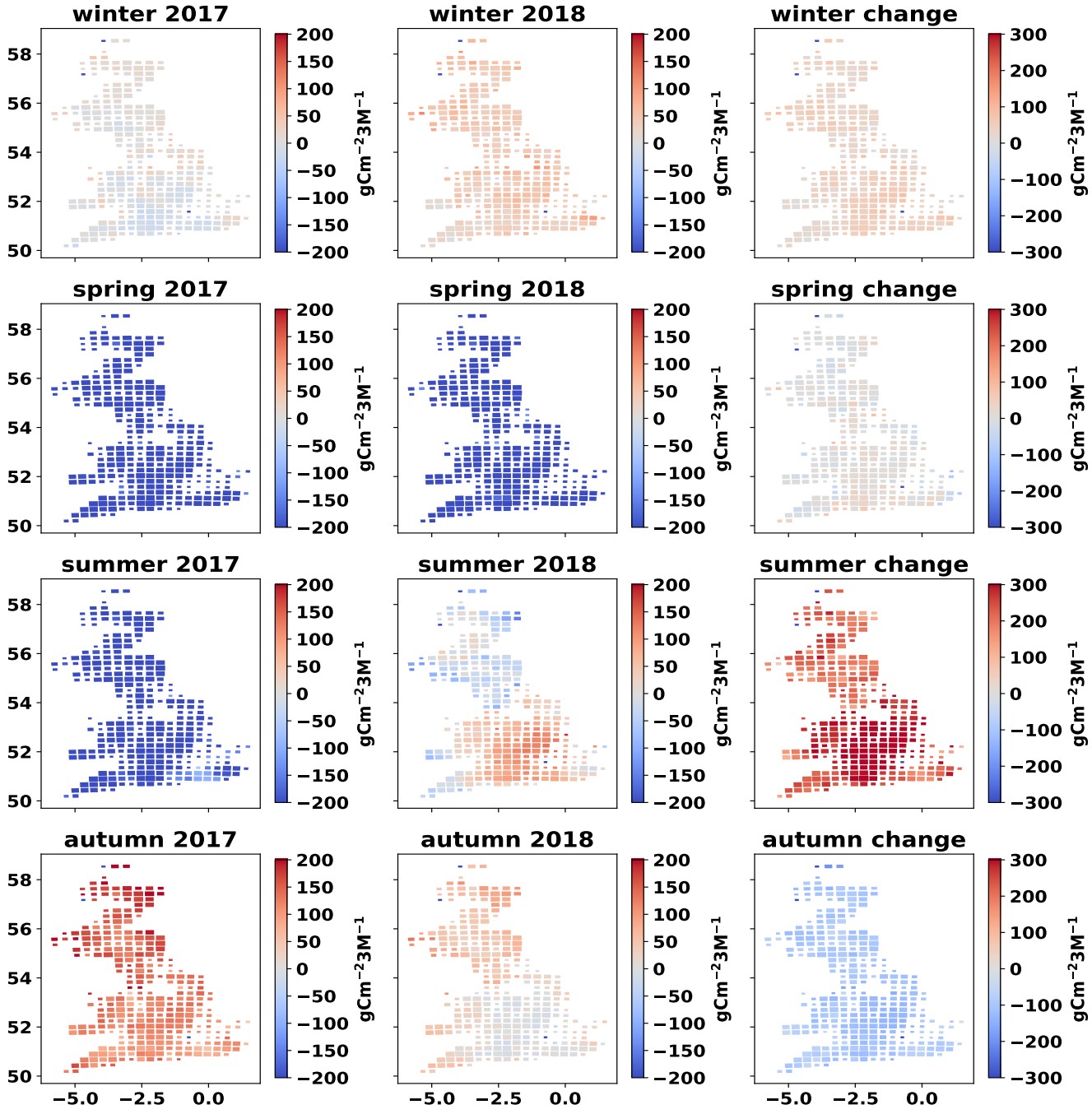

**Figure A8.** Cartograms of cumulative NEE per season for 2017 (December 2016 - November 2017) and 2018 (December 2017 - November 2018), and change in seasonal NEE from 2017 to 2018. The mean MDF-predicted seasonal NEE of all fields in each cell is presented. The size of cells is adjusted according to the number of simulated fields within it.