# Peer review of "The carbon budget of the managed grasslands of Great Britain - informed by earth observations"

_Biogeosciences, 2021_

## Author Response (AR1)

**Authors response**

We would like to thank the three reviewers and the editor for their constructive comments. Here, we describe how we responded to each comment and point to the relevant changes in the revised submitted manuscript. The manuscript has undergone many changes. Our introduction is now more focused on the study's advancements/novelties which are reiterated in the discussion and conclusions. Materials and methods have been revised in depth to ensure our methodology and the use of data and modelling are clear to the reader. Due to the extensive nature of the revisions made to the manuscript we occasionally refer to entire sections rather than specific lines and pages.

**Referee #1 (Anonymous)**

> *.... However, the current description of the Materials and methods are not clear, which prevent making further assessment of the quality of this study.*
> We have made major and extensive revisions to the materials and methods section of the manuscript. We believe that our revisions sufficiently address this important issue raised by all three reviewers. We point to the entire MM section for the changes made.

> *It is not clear why and how the two sets of remote-sensed LAI were used in the study. I would think both are EO-based data. It seems that CGLS LAI were used as input to DALEC- Grass, while Sentinel-2 LAI were used in CARDAMOM to optimize parameters of the DALEC- Grass model. As one set of LAI was used as input, it will not be surprise the modelling framework can give reasonable LAI against LAI from another dataset. I would suggest the authors to clarify the reason and the necessity of using the two sets of EO-based data..... After going through the Materials and methods section, it is still not clear e.g., how the different models/frameworks were connected; how and where the EO-based data were used; how the model parameters were optimized; how the C fluxes were estimated. A flowchart represents all the inputs, model connections, and outputs with step-by-step procedures will be very helpful.*
> We clarify these issues in the revised manuscript by (1) adding a dedicated section to MM (section 2.2.2), (2) revising section 2.1.2 and 2.1.3. (3) revising section 2.1.4 and justifying the use of two EO datasets (line 202 page 7) (4) providing schematics on how EO data and the model are combined (fig A1 appendix) and (5) adding a new section (2.2.2) and schematic of how grazing and cutting are inferred and calculated (Fig 3 page 10)

> *It is not clear how grazed/cut events (as a critical result of this study and an important component of the C budget) were identified, and grazed/cut biomass was simulated. These are the most interesting and important part of this study. This part of the methodology may need more details. An example of i) the variations of original EO-based LAI, modelled LAI, associated C fluxes, ii) how exactly the grazed/cut events were identified, iii) how "mostly grazed" and "mostly cut grasslands" were differentiated is necessary.*
> We added a section (2.2.2) dedicated to this issue and provide a schematic on how grazing and cutting are inferred and calculated (Fig 3 page 10) in the revised manuscript

*> The components of C budget were only very briefly mentioned. It is not clear how each component was estimated. Especially for manure, I can not find how it was estimated (or derived from another dataset).*

> Information on how manure and other livestock-based fluxes are calculated has been added to section 2.1.2 (line 150 page 5). We clarify how we calculate the C budget/balance in a new dedicated section (2.2.3).

*> it seems that the manuscript was not carefully checked before submission. There are plenty of "i.e. …"," ,"?"??" in the text (e.g., L80, L91, L175, L209, L231, L298, L374 etc.) that looks like unsolved comments from the authors, and the manuscript is not taken seriously at all.*

> We would like to apologise again for the presence of "?" In the initial submission. These were not unanswered questions but show the location of references that the software used to produce the submitted .pdf could not find (unexpectedly)

**Specific comments :**

- *How the sampling of grassland fields can result in only 1-5 simulated fields per cell? What are the Metropolis-Hastings (MH) method and the Simulated Annealing (SA) algorithm? What is the difference between them.* We removed this reference to the number of fields per cell because it is confusing. When we referred to cells in the cartograms we referred to the cartogram cells (that grow/shrink depending on number of fields within) and not the cells of the 25km grid that we used to sample the fields. We also removed the reference to MH as this is already discussed in detail in previous papers and it would require extensive new text to describe in the manuscript.

- *For Fig. 2 and 4, it will be very useful to show not only the absolute values/biases, but also fraction of bias or (mean of MDF-predicted – census) / census, and maybe discuss the reason of bias. For Fig. 4, it might give insights on the mismatch due to the different years of prediction and census.* We are not sure what is required here. We argue that the numbers (section 3.1) of figures (Fig A2 appendix) provided give sufficient information to the reader to understand how effective the assimilation of EO LAI time series was.

- *How the mean C fluxes across the GB were calculated? Area weighted? If so, how? Whether the selected points are representative for all grassland grid cells?* The simple (not area-weighted) mean is presented in tables. What is presented in the results in cartograms is the simple (not area-weighted) mean of all fields within each cartogram cell.

- *It would be necessary to provide the maps of rough grazing, permanent and temporary grassland, and the maps of resulted management type (e.g., grazed only field or grazed + cut field), grazed, and cut biomass for users to understand the management intensity.* Such maps would have been very useful but, unfortunately, do not exist.

- *It is strange that NEE/NBE were negatively related to both GPP and REco.* Please note that the micrometeorological sign convention is used to NEE and NBE (see section 2.2.3 in revised MS). Higher GPP (and thus higher Reco also) is related to more C sinking activity and, therefore, to lower NEE/NBE.

- *As the uncertainty for LAI is nearly half of mean LAI, the robustness of the prediction should be further discussed.* We added new text in the revised MS on this issue (section 4.4. page 22)

- *"Mostly grazed" and "mostly cut grasslands" were not explained before results section.* We clarify this in advance (section 2.2.3, page 10, line 254) and clarify its meaning when used (i.e. figure caption)

- *Paragraph started from L409: It seems that the second assumption is not an assumption but observed phenomena. The logic of the discussion here is hard to understand. Why the C source/smaller sink caused by drought in 2018 can infer management is more important than climate?* This paragraph was indeed confusing and has now been revised (line 464, page 21)

- *All the abbreviations will need to be explained in the main text in addition to the "Abbreviations" in the beginning.* This is now done in the revised MS in the introduction and the MM (section 2.2.3)

**Referee #2  (Aiming Qi)**

> *it may be more proper to replace the "constrained" used in the title "The carbon budget of the managed grasslands of Great Britain constrained by earth observations" with "adjusted" or "estimated".*
> We have revised the title of the manuscript to "The carbon budget of the managed grasslands of Great Britain - informed by earth observations"

> *What were included in the managed grasslands? Did they include rough-grazing grasslands in the context of three UK grassland types – temporary, permanent and rough- grazing ?*
> The simulated fields were sampled by using a land cover map to randomly sample 1 field per 25km from across Great Britain. Therefore all three types of grasslands (rough-grazing, temporary, permanent) are, in theory, included in the sampled set of simulated fields. Based on the fact that rough-grazing and temporary grasslands cover ~90% of all grassland area, while temporary grasslands cover the remaining ~10, we believe that most of the simulated fields are rough-grazing and temporary. We cannot know in advance if a simulated field is rough-grazing or temporary but we can infer what it is based on the estimated total annual grass biomass yield. This is what we do in sections 3.2 and 4.1 of the revised manuscript

> *It was said that there were 1855 fields selected for simulations across GB in 2017 and 2018. How many fields were selected in 2017 and 2018, respectively? How many fields were grazed only, how many fields were cut only and how many fields were both grazed and cut? What were the total areas for 1855 fields and in each management grassland type? It would be good to make a box plot showing the size distribution of selected 1855 fields.*
> The same 1855 fields were simulated for 2017 and 2018. Section 3.2 (Line:318, Page:13) clarifies how many fields where grazed, cut, grazed-and-cut. Due to the spatial resolution of one of the two earth observation datasets used in the study we had to impose a filter on the size (6-13ha) of the sampled fields to-be-simulated (process of sampling described in section 2.2.1 Page 8).

> *When selecting fields to be included, the passing criterion was 50% overlap limit. What did the overlap measure specifically? It was also necessary to know how many fields were ignored when simulations were compared with LAI from EO data.*

> Overlap quantifies the % of EO-based data (field-mean LAI) that are within the corresponding MDF-predicted 95% confidence intervals (Line:270, Page:11). The number of fields that failed to pass the 50% overlap limit is stated in the results (Line:300, Page:12)

> *The manuscript was not cleanly finalised before it was submitted to the journal website because there were many places that had unanswered question marks in the manuscript.*

> We would like to apologise again for the presence of "?" In the initial submission. These were not unanswered questions but showed the location of references that the software used to produce the submitted .pdf could not find (unexpectedly)

> *Flow of information between models used in the coupled MDF algorithm framework was not clearly presented. So, an added diagram may be helpful.*

> This is a major concern that is shared among the reviewers. We have made extensive revisions to address this issue in the materials and methods section. Figure 1 was updated, Figure 3 was introduced to describe how cutting and grazing are inferred, Figure A1 (appendix) shows the data used, and the flow of data in/out of the model within model-data fusion algorithm/framework.

> *The "Removed biomass" item in Table 1 was 220 in 2017 and 280 in 2018. If 2018 was extremely hot and dry summer, why was there more biomass for removal because of limited pasture herbage yields? What was included in the "Removed biomass"?*

> An explanation on why the mean removed biomass (grazed and cut) was higher in 2018 compared to 2017 in line 360 page 14. Removed biomass includes all grazed and cut biomass within a year (note added to table 1 page 16)

*Specific points raised by the reviewer :*

All the points raised by the reviewer were dealt with in the revised document. Most of them were missing references and typos. The following points need authors' response :

1. *Livestock Unit (LSU). It is more customary in the UK that "LU" is short for livestock unit.* : Indeed, we are now using LU in the revised manuscript
2. *"21-day average photoperiod(sec)". When was the starting date from which the 21 days were counted?* : This is a 21-day rolling average. Calculation starts from simulated day 1 i.e. 1-1-2017. We believe this issue does not require the addition of relevant text in the manuscript
3. *The agricultural census data for England was in 2010. The LAI from EO data was in 2017 and 2018. The temporary grasslands must have been changed into other land use types during these 7-8 years gaps. So, the grassland supporting animal number statistics cannot be accurately compared between the two time points.* : The 50% overlap limit ensures we are only simulating grassland fields. Indeed, animal statistics are collected at the level of local administrative units and for England the more recent available data are from 2010. We know that there has been a decline in livestock numbers since 2010 across the UK/GB. This explains the small negative bias in predicted LU ha-1 (section 3.2 page 12). However, when we look at things at the national/GB scale —by examining/analysing ~2000 fields from across GB— we can robustly compare the predicted and recorded national-scale spatial distribution of livestock (Fig. 4). We can, therefore, credibly answer the question of whether we can

track the relative distribution of livestock across GB. The only way to "accurately" quantify livestock units on individual grassland fields is by obtaining the actual number/type of livestock from the farmer. Agricultural census data are not collected/provided regularly and refer to local admin units. Therefore, census data do not provide accurate information for individual fields also. We show that if/when we want to obtain field-specific estimates of livestock numbers, grazing/cutting patterns across larger spatial scales our method is effective. The uncertainty of the relevant estimates represents the "cost" for obtaining these estimates. Reducing this uncertainty could be achieved by further developing/testing the MDF algorithm and the underlying EO data processing routines.

4. ***There were many types of sheep. The 0.11LU is a sheep. What was the sheep used here, lowland sheep or highland sheep? 70kg or 80kg sheep?*** **:** We provide a reference to LU calculation in line 276 page 11 of the revised manuscript. As we cannot infer neither the type nor the age/weight of animals from satellite EO data we have to rely on simplifications and on our knowledge on the main types of livestock in the UK (i.e. dairy/beef cattle and sheep) to calculate predicted LUs. When considering our calculation-of/results-on field-specific LUs one should take into account that there is no alternative method for estimating livestock type and density in individual grassland fields across large spatial domains — excluding farmer information provided directly (thus a very limited number of fields/farms) and not field-specific spatially-aggregated agricultural census data (available every ~5-10 years).

5. ***"The MDF-predicted GB-average pasture dry matter yield (6±1.8 tDMha-1y-1)". Was this referred to 2017 or 2018 or in both years? It was for both years, can values be given for each year, too?*** **:** Yes, this refers to both simulated years. Values are given for each year (Table 1 page 16)

**Referee #3 (Community Comments)**

> ***.... The manuscript is worth to be published but would need some clarification in the MM section to help the reader to get through.***
> As stated and described in our response to relevant comments by the other two reviewers we have made extensive revisions to the MM section.

> ***I was wondering if a flow chart /scheme would help to guide the reader though the "model simulation"; i.e allowing to distinguish between "hard/real" data inputs from databases, (soil grid, management practices Edina AgCenesus and meteo) and those which are "elaborated" EO LAI data and how the feed into each other...***
> We are providing a schematic (Fig A1, Appendix) on how data (EO, weather, soil C) and modelling are used in this study in the revised MS

> ***Along the manuscript I missed some explanation on the difference between C sequestration and NBE??***
> In the revised MS we use a new dedicated section to clarify what is calculated and presented (section 2.2.3)

> ***As well as how to get from one term to the other ect. (E.g. Why harvest is not removed in the NBE Table1, L371ff), as to my understanding NBE= NEE-harvest+manure. In short, C balance, used terms and NBE vs. SOC changes (C Sequestration), needs clarification in MM and not only in the abbreviations!.***

> Table 1 shows the mean values (I.e. mean NBE NEE removed biomass) for each year across all simulated fields. Removed biomass varies significantly across the ~2000 simulated fields with grazing (thus manure returns) being more prevalent than cutting. Therefore, a simple addition of annual NEE and annual removed biomass values in Table 1 should not be expected to be equal to the corresponding annual NBE. We clarify what NBE and Δsoc quantify/present in section 2.2.3

> ***The same was for manure. Where did Manure come from and how Cardamom accounted for Manure (C/and N) ? EDINA database? (see L368)***
> This is now clarified in Fig 1, page 6 and the revised MS section 2.1.2

> ***Personally I would add the section of Limitation of the approach and opportunities (e.g. SWOT) here. As well as suggestions, what can be modified and what we can learn? However, these sections are standalone at the end of the manuscript, and I wonder if they should/can be moved to the corresponding sections (at the top of the discussion instead of 4.4 uncertainty and 4.5 limitation), which would make them more complete for the reader. Having said this, the section on C balances need more discussion on the limits of the study, the usefulness for national inventories (i.e. NBE vs. SOC changes see section future work), ....***
> Our limitations section (4.5) has been revised and enhanced with additional discussion of how to not mis-interpret results and what is/is-not considered by DALEC-Grass and the MDF algorithm.

> ***Specific comments :***

> ***L 87 may be cite : Pique et al 2020 Remote Sens. 2020, 12, 2967; doi:10.3390/ rs12182967 :*** This is a very interesting and relevant reference/study and is cited in the revised MS

> ***L 134ff: "At each time step the algorithm reads the vegetation reduction information and decides whether to simulate the corresponding ... " this is not quite clear and I wonder of a flowchart will help? What is the time step? :*** This is clarified in the revised MS through new sections (2.2.2) and text revisions (in sections 2.1.2, 2.1.3)

> ***L212ff "To assess the effectiveness of the LAI assimilation process we quantify the level of fit between MDF-predicted and EO-based time-series using ..." Until now I did not get that Cardamom estimates LAI (see L 141 and L241) put is used this as an input. Seems I have missed a point. Can authors please clarify. (eg in a scheme?)*** : Schematics in fig 3 page 10 and Fig A1 clarify this in the revised MS

> ***L234 "The estimated SHAP values are normalised (0-1) to be comparable to r2 ." So 1 would be very good ? and what is the number for low fitting (ie limit of SHAP) :*** This means that a RF predictor with SHAP = 1 explains 100% of the variance in estimated annual NBE (i.e. ~ corresponds to a r2=1)

> ***L235 and L 331ff "RCR is equal to the size of the MDF-predicted 95% confidence interval divided by the corresponding..." please help the reader to get the number in the right way. eg RCR is 42 ± 9% for LAI, means the uncertainty of LAI is 43% so very high? Or very low? With respect to which best value? :*** The revised MS section 4.4 describes

how the reader should look at the estimated predictive uncertainty by using the observational uncertainty as a benchmark

> **I am not quite sure the cited studies (L390ff) were interpreted in the right direction. :**This paragraph has been revised because it was, indeed, confusing (line 555 page 24)

> *L416 "conclude that management is more important than climate in terms of the C balance of managed grasslands in GB."  --do authors have a citation which confirm/ underline this interpretation :* This statement has been revised (see line 566 page 24)

---

## Referee Report (RR1)

Thanks for the very nice revision! From my point of view, the authors well addressed the comments from the other two reviewers and me, and the manuscript is now in a good shape. I believe this study is very valuable to the community. Here are some of my remaining minor remarks:

L273: the "()" is redundant.

L400: please double check the usage of GCD<0 and GCD>0 in this section. For example, GCD<0 sometimes means mostly-grazed, but means mostly-cut in other cases.

L421: Hr should be Rh.

Table 1: The NBE can not be obtained from the values presented in the table. It would be helpful to give values for all the components of NBE and ΔSOC. In addition, it is not clear what is the meaning of C flux into soil. Does it include litter and manure?

L434: It is not clear what are included in the "high inputs of C to soils", litter + manure? Does manure from refinement included in this study? If not, it should be mentioned and discussed. Because it will cause an underestimation of C input for grassland.

L600: I would think it would be helpful to use the meaningful parameters' name (e.g., PNUE, or LCA) rather than the Code of parameters (e.g., P10, and P15) across the manuscript.

---

## Author Response (AR2)

**Authors' response**

We would like to thank the two reviewers for their constructive comments. The location of changes made to the manuscript is stated in this response and can be seen in the submitted document (with tracked-changes). In addition to this response and the revisions made to the manuscript we also intend to submit an article to the EGU Biogeosciences division blog. This article will plainly describe our methodology and discuss the role of earth observation for grassland management inference and ecosystem biogeochemistry modelling.

**First reviewer**

[**Comment #2**] *L400: please double check the usage of GCD<0 and GCD>0 in this section. For example, GCD<0 sometimes means mostly-grazed, but means mostly-cut in other cases.*

[**Response to comment #2**] We ensured GCD is used correctly in the text

[**Comment #4**] *Table 1: The NBE can not be obtained from the values presented in the table. It would be helpful to give values for all the components of NBE and ΔSOC. In addition, it is not clear what is the meaning of C flux into soil. Does it include litter and manure?*

[**Response to comment #4**] Table 1 does not present the total/sum annual NBE/NEE but the mean across the ~2000 simulated individual fields (i.e. area average). Also, the estimates for each simulated field show the mean predicted since our model-data fusion framework is probabilistic. For these reasons, one should not expect the area-average annual NBE to be equal to : the area-average annual NEE + Bc + Bg - Manure. We have added the area-mean (+\- SD) predicted value for manure-C to Table 1. "C flux into soil" has now been removed from Table 1. The term was used in our initial submission to describe the C that flows from the litter to the SOC pool but was replaced with Δsoc, which is more informative and easy to understand.

[**Comment #5**] *L434: It is not clear what are included in the "high inputs of C to soils", litter + manure? Does manure from refinement included in this study? If not, it should be mentioned and discussed. Because it will cause an underestimation of C input for grassland.*

**[Response to comment #5]** Figure 1 shows the C pools and fluxes simulated by the model. Manure from refinement cannot be inferred from earth observation data and relevant data (agricultural stats/census etc) cannot be spatially disaggregated in robust ways. As described in section 2.1.2 of the MS, at every time-step (i.e. week) manure is simulated as being produced by grazing animals in proportion to the simulated grass consumed. The grazing livestock-produced manure is immediately deposited to the soil (i.e. enters the soil litter pool). We discuss the fact that simulated manure production/deposition is based on inferred livestock density in the limitations section (4.5) of the MS.

**[Comment #6]** L600: I would think it would be helpful to use the meaningful parameters' name (e.g., PNUE, or LCA) rather than the Code of parameters (e.g., P10, and P15) across the manuscript.

**[Response to comment #6]** Some model parameters have very large names that cannot be abbreviated. We follow a convention when referring to model parameters the use of parameter codes is preferred because it helps us avoid using very long sentences at certain parts of the MS. We understand, however, that having to look at Table A1 is not easy for the reader, this is why we use abbreviations for those 2-3 that are frequently mentioned in the MS.

**Second reviewer**

**[Comment #1]** *Net Carbon flux - My main concern is that the manuscript does nothing to convince the reader that the net carbon fluxes (NEE and NBP) can be inferred by assimilating only leaf area index (LAI) data. This outcome seems counter-intuitive. I can accept that assimilating LAI can provide better estimates of GPP and possibly Ra. However, it is far from clear that this will give the correct results for Rh and hence NEE or NBP. I understand Rh in the model to be driven primarily by a temperature response and the amount of soil carbon. If I have understood the manuscript correctly the soil organic carbon is set by using data from the SoilGrids data base and the initial value is not tuned as part of the data assimilation. So, in essence, this analysis is attempting to improve the temperature response of Rh based only on observations of LAI. There is an "EDC" that constrains the rate of change of the SOC pool (EDC #3), which is not unreasonable, but I am not convinced this necessarily helps get the values of the parameters that control Rh correct. The main thrust of the paper is the carbon budget of GB grasslands, so I think it is incumbent on the authors to provide some evaluation, otherwise it is really only model*

*output with no indication of how trustworthy it is. I have skimmed the two cited publications by the lead author on this subject and, as far as I can tell, the only comparison with NEE is at a single site (Easter Bush). Also, in that study, the methodology had notable differences from the current one (no EO data, different list of EDCs and so on). Some validation of the net fluxes is required.As a minor point in reference to the above, I also notice that EDC #2 constrains the size of the SOC pool (Table A2) but the initial size of the SOC pool is not one of things tuned, according to Table A1. Presumably this EDC isn't used in assimilation in this study?*

**[Response to comment #1]** This is the main comment of the second reviewer and we would like to provide a thorough response.

*My main concern is that the manuscript does nothing to convince the reader that the net carbon fluxes (NEE and NBP) can be inferred by assimilating only leaf area index (LAI) data. This outcome seems counter-intuitive. I can accept that assimilating LAI can provide better estimates of GPP and possibly Ra. However, it is far from clear that this will give the correct results for Rh and hence NEE or NBP*

We argue that a quantitative study, which focuses on a specific type of ecosystem in order to provide estimates at high resolution (spatial/temporal) and across a large domain, cannot be validated against flux data just as a field or landscape scale study can. This is because there are no measured C flux data to compare predictions with at the pseudo-national scale. We clearly state and highlight in the MS that the credibility of model estimates, at the resolution/scale of our study, depends on (1) model calibration/validation within the domain of application; and on (2) whether or not observations are used to —even partly— validate model predictions. In this respect, we have used two datasets to calibrate and validate the DALEC-Grass model: (i) the most extensive ground-measured, managed grassland-specific dataset of C pools and fluxes available in the UK (Easter Bush site) and (ii) a shorter measurements dataset produced by using different state-of-the-art $CO_2$ measuring instruments (Crichton site). Based on this fact, we argue that we are using a calibrated and validated model whose parameter priors reflect the biogeochemistry of a typical UK managed grassland (dominated by perennial ryegrass, with some clover, that has been a grassland for years/decades). In terms of the use of observations, this study is the first in the UK that uses observational data on a key aspect of grassland C cycling (aboveground biomass volume) for the purposes of quantifying C pools and fluxes. Because of that, we argue that this study produces more credible results than previous, relevant quantitative studies.

*If I have understood the manuscript correctly the soil organic carbon is set by using data from the SoilGrids data base and the initial value is not tuned as part of the data assimilation.*

We would like to thank the reviewer for pointing out that the initial SOC pool size parameter was missing from Table A1 (now added). The size of the soil organic carbon (SOC) pool of every simulated field is an optimisable model parameter. The prior range of SOC ranges between  +/- 10% of the spatially-corresponding SoilGrids value. Using SoilGrids data to set an initial value for each field's SOC pool allows us to control the model's predictive uncertainty. This is important because the size of the SOC pool is the largest source of uncertainty around grassland C cycling estimates.

*I have skimmed the two cited publications by the lead author on this subject and, as far as I can tell, the only comparison with NEE is at a single site (Easter Bush). Also, in that study, the methodology had notable differences from the current one (no EO data, different list of EDCs and so on). Some validation of the net fluxes is required.*

In the first of these two cited publications we have used 11 years of daily-measured data from two variably-managed grassland sites in Scotland, UK in order to refine the parameter priors and validate the predictions of DALEC-Grass. This data included : soil surface respiration, above and below-ground biomass, ground-measured leaf area index and chamber and eddy-covariance-based NEE measurements. Beyond the Europe/grassland-focused studies already cited in the MS we do not know of other relevant recent studies that provide measured/modelled grassland net C flux estimates, and which could be used for further validation of our results. We would be glad to include more observational studies/data on UK grassland NEE if this reviewer can point them out. Considering the uncertainty around field-measured C flux data, it is generally believed that permanent grasslands in the UK (and NW Europe in general) are almost C neutral (i.e. NEE = ~0). The results of our study are in agreement with this statement. We would like to highlight again the fact that, in contrast to the majority of model-based studies on grassland C fluxes at large scale, our predictions are not "completely unvalidated" as observational LAI data are assimilated and thus estimated aboveground biomass/C is being validated. Moreover, we are particularly interested in seeing studies discussing/presenting the impact of the 2018 heatwave on grassland NEE. We have cited a number of studies that do this using model predictions and measured point data extrapolations. Unfortunately, we could not find any UK measurements-based studies discussing the impact of the 2018 heatwave. We anticipate

such studies to be published soon. In this regard, data from monitored managed grasslands in the UK show the positive response in NEE (reduction in C sinking) that our simulations are predicting (http://nora.nerc.ac.uk/id/eprint/525106/1/N525106PO.pdf)

[**Comment #2**] *Use of EO LAI - Despite requests from previous reviewers it is still not clear how the two different EO LAI data sets are used, and the justification for using them both is not well made. I read the relevant sections several times and it is still not clear. It appears that CGLS data are used as a driver to quantify "vegetation reduction" and the Sentinel-2 data is used for assimilation. I suggest a complete rewrite of these sections to include a much clearer description of how this is done. In addition the use of the GCLS data is not well justified. The argument seems to be that it is too spatially coarse to represent a field, but have sufficient spatial resolution to detect grazing or cutting. I personally do not understand this. The choice is apparently driven by a better temporal resolution (10-days) than Sentinel-2, but the combined Sentinel-2 instruments actually have a shorter revisit time than this, so the only advantage appears to be that the GCLS data are gap-filled. But (a) won't the gap filling itself reduce the ability of the data to represent grazing/cutting? and (b) why not gap fill the Sentinel-2 LAI data? Can the authors provide a better justification for using both data sets?*

[**Response to comment #2**] Indeed the main reason for using the CGLS data is the 10-day temporal resolution. This is important when considering that there are ~25 cloud free Sentinel2-based images per year; and even fewer images in coastal UK areas. However, the temporal resolution of the satellite-based data is not the only reason for using the CGLS data. Firstly, CGLS data are produced using images retrieved by a different satellite system (originally Proba-V and since more recently Proba-V + Sentinel-3). This means that we constrain the model-estimated LAI (thus aboveground biomass/C) using information from two different systems, which is, per se, more robust than relying on a single system. Secondly, we agree that we could have gap-filled cloud-free Sentinel-2-based LAI data points to produce continuous LAI time series. However, had we gone down that road we would have developed and tested a method very similar to that used by CGLS. This is because, grassland vegetation volume changes within a year in ways that are much less predictable and visible than e.g. crop and timber harvesting. Therefore, the "best" way to interpolate between scarce LAI data points is to use past/historical and/or neighbouring grassland-specific pixel data; which is the method used to produce the CGLS data. Moreover, relying on the freely-available, well-documented, and continuously-maintained and updated CGLS data is better than using any in-house and partly-validated method/data.

We believe that we have extensively revised the MS to explain how and why we are using the CGLS data in this study in our previous revision. This revision included adding Figure 3, which we believe clarifies how CGLS and Sentinel-2 data are used (when/where they are used). We cannot see how re-revising the relevant text can further clarify things.

We would like to add at this point that we see the spatial and temporal resolution of the EO data as critical to the accuracy of the predictions of the model-data fusion algorithm. For this reason we are working on developing a robust, grasslands-tailored and reproducible method to interpolate Sentinel-2 based vegetation indices (LAI in particular). This is still work in progress but our initial testing shows that we will be able to stop using the CGLS data in the near future. This work is pending further validation using a larger ground-truthing dataset (see https://datashare.ed.ac.uk/handle/10283/4086 for more details).

In conclusion, it is not unreasonable for the reader to wonder why we are using the EO data product that we use. However, the main arguments for using the CGLS data in addition to Sentinel-2 data (i.e. temporal resolution, validation and maintenance of the CGLS data) are presented in the MS. We do not think that dedicating more text on EO data choices is necessary; especially when considering (1) that our response to reviewer comments is public, (2) that we intend to further discuss EO data for grasslands in a blog article and (3) that our previous recent publication presents/discusses the pros/cons/effectiveness of using the CGLS data.

[**Comment #3**] *SHAP values - This is a more minor point than the previous ones, but the Random Forest approach appears to have been used solely for the purpose of obtaining SHAP values. A potential issue with this is that the SHAP values tell us about the sensitivity of the machine learning model to its feature space and not necessarily about the mechanistic model. Consequently, it can result in misleading conclusions if one is trying to infer things about the model that has been emulated, for example when two or more features are correlated. There are other techniques that work directly on models to perform sensitivity analyses and given the model used in this paper is sufficiently computationally efficient to perform MCMC calibration, it would seem an odd choice to emulate it just to back out these sensitivities. Furthermore, the fact that the correlation analysis (Fig 7) provides very similar information tell us it has not added a great deal to the analysis. Given that, I suggest removing the parts about SHAP values.*

[**Response to comment #3**] Indeed because DALEC-Grass is mechanistic we can explain its behaviour and the logic behind its predictions. In general, the SHAP method is used here in the same way that the correlation analysis is.  However, we believe that building a machine

learning (ML) model and using SHAP to present its sensitivities is interesting and has to be included because SHAP offers a quantitative assessment that is clearer compared to that of the correlation analysis. We also believe that this RF + SHAP section of the MS provides a brief test using ML + SHAP as a method for creating an emulator, quantifying its predictive ability (R2) and its sensitivities. We believe that ML + SHAP has the potential to be used to assess and apply an ML emulator in order to extrapolate site-scale model-data fusion-based GHG flux estimates. While this is beyond the scope of our study and do not discuss it we believe that keeping the RF + SHAP section in the MS will be useful small addition to the relevant literature.

**[Comment #4]**

*L170: Do you really mean that you calculate the likelihood from the RMSE?* Yes, RMSE is used as the log likelihood.

*L190: The sentence here is a bit odd. I don't understand why it's relevant to state that the data are processed from top-of-atmosphere reflectance. Presumably they are corrected to surface reflectances prior to estimating LAI?* Indeed, we have reworded this sentence in the revised MS.

---

## Author Response (AR3)

We would like to thank the editor for helping us to improve this manuscript (MS). We apologize for failing to see that the tracked changes .pdf in our last submission had its last pages missing. Our MS copy is produced using LaTex. A conflict between *latexdiff* (needed to track changes) and the official supplementary files of the Copernicus LaTex package led to an incomplete .pdf being produced. Now our tracked changes document has been produced in a different way. Below we refer to each of the reviewer and editors comments, point to the MS location where revisions were made and provide a written response. We use the page (P) and line (L) number in the revised MS when pointing to revisions made.

**1. Please clarify the seemingly interchangeable use of the terms GB and UK**

**Response :** We have clarified this in the revised MS and now use GB throughout.

**2. Comment #4 of reviewer #1 (for the second review of the MS)**

*The NBE can not be obtained from the values presented in the table. It would be helpful to give values for all the components of NBE and ΔSOC. In addition, it is not clear what is the meaning of C flux into soil. Does it include litter and manure?*

*Editor comment : But this is not what the table caption says, please revise caption*

**Authors response :** We understand that this table has been the source of confusion regarding what is and what is not included in the presented variables. We replaced the table with a clearer schematic of estimated C pools, fluxes and balance (Figure 6, P18). The new figure shows the simulated area-mean (and standard deviation across all fields) of C pools and fluxes for the two simulated years (2017 and 2018); as well as the GB-mean annual NEE and NBE. The figure clarifies what was simulated and compares the size of pools and fluxes during the two simulated years.

**3. Comment #5 of reviewer #1 (for the second review of the MS)**

*It is not clear what are included in the "high inputs of C to soils", litter + manure? Does manure from refinement included in this study? If not, it should be mentioned and discussed. Because it will cause an underestimation of C input for grassland.*

*Editor comment : The fact that manure cannot be inferred from EO-data or inferred from land-use statistics, but is an important component of the C-balance is one of the important reasons why there needs to be more care when going from the implications of your study from effects on LAI and GPP to the whole ecosystem carbon balance. I don't see this properly explained in the tracked-changed version of the manuscript.*

**Authors response :** Unfortunately some of the text that was added in response to this comment (after the 2nd review) was in the limitations section of the revised MS; which was the part that was missing from the last submitted tracked changes pdf. Reference to the fact

that external manure additions are not considered in this study, and the likely implications, can be found in 4 different parts of the revised MS :

(1)   abstract (L14)

(2)   materials and methods section 2.1.2 (L177)

(3)   materials and methods section 2.1.5 (L225)

(4)   Discussion of limitations (L616).

The addition of manure to the soil is an important component of the C balance of any grassland. In this study, we assume that the amount of manure added to a field's soil equals that produced by the livestock that graze it and that it is applied weekly and directly after each grazing even (i.e. biomass-C is grazed and part of it returns as manure-C to the soil). This application mode is a simplification. In reality livestock spend time indoors (mostly in winter, autumn). During these "housing" periods, livestock excrements are collected and stored to be either applied later on a farm's fields or sold to other grassland and/or arable farms. It is not possible to know how much manure is applied to each field but it is credible to assume that manure production and application is closely related to and driven by farm livestock density. This is particularly important when looking at things at GB scale. We refer to this in section 4.5 (L616-627) of the revised MS and added a citation to the relevant literature.

**4. Comment #1 of reviewer #2  (for the second review of the MS)**

*My main concern is that the manuscript does nothing to convince the reader that the net carbon fluxes (NEE and NBP) can be inferred by assimilating only leaf area index (LAI) data. This outcome seems counter-intuitive. I can accept that assimilating LAI can provide better estimates of GPP and possibly Ra. However, it is far from clear that this will give the correct results for Rh and hence NEE or NBP*

*Editor comment : I take all your points, but I fail to see how this discussion has entered the discussion section of the manuscript (the tracked changed version ends prematurely at page 22). I also disagree that a random error on the SOC initialization will result in an appropriate quantification of the SOC related error, since soil grids does not (to my knowledge) use grazing versus cutting as a factor in the upscaling. Given that you find notable GPP differences between these types of management, one might also expect differences in soil C, which isn't/cannot be captured by your approach. I also agree that there are limited C balance data for UK and other grasslands, but again, this is a caveat that needs to be clearly stated. I agree with your statement that your study is an important improvement over earlier studies not taking account EO data, but, as reviewer #2 pointed out - it is far from yielding a comprehensive C balance for UK-grasslands. This limitation is also not appropriately reflected in the abstract, where the reader is not informed that the EO-informed grassland growth model is then used to prognostically simulate the carbon balance of UK-grasslands, with a range of important yet unconstrained terms in the calculation.*

**Authors response :** We clarify that our diagnostics are based on probabilistically implementing a model that has been calibrated and validated against what we believe are the most detailed and extensive set of relevant ground measured data available in GB [1]. We do not infer exact C pools, fluxes and net C balance (ecosystem/biome) simply by assimilating LAI time-series. Instead we use EO-based LAI time series to constrain model estimates of aboveground biomass dynamics and use the modelling of underlying biogeochemical processes to generate probabilistic estimates of NEE. The modelling is supported by the setting of realistic priors on parameters through the detailed calibration undertaken in earlier studies using appropriate relevant ground data [1,2]. We discuss this it in the introduction (L121) of the revised MS. Also we added a reminder/note of/to the approach in the discussion section (L490).

In relation to the soil C model input data, we failed to make it clear that, over decadal periods relevant to soil C dynamics, the majority of GB grasslands undergo alternations between cutting and grazing and so their management varies from year to year (relevant comment was added to L525). Therefore, there is no fixed spatial pattern in defoliation practices across GB and no certainty over which practices are applied each year. This is why we rely on EO to infer defoliation methods at field scale. Our study quantifies how grazing/cutting management affects biomass productivity (GPP) and litter inputs. The initial model soil C stocks are constrained with SoilGrids data and then soil C dynamics depend on simulated inputs and outputs. We note that spatial information on vegetation type (land cover maps) and biomass productivity (satellite data) are among the factors used to go from national-domain ground-measured point-based datasets to the global-scale gridded data that are provided by SoilGrids.

[1] Myrgiotis et al. A model-data fusion approach to analyse carbon dynamics in managed grasslands. Agricultural Systems 184, 102907 (2020). https://doi.org/10.1016/j.agsy.2020.102907

[2] 1.Myrgiotis, V., Harris, P., Revill, A., Sint, H. & Williams, M. Inferring management and predicting sub-field scale C dynamics in UK grasslands using biogeochemical modelling and satellite-derived leaf area data. *Agricultural and Forest Meteorol* **307**, 108466 (2021). https://doi.org/10.1016/j.agrformet.2021.108466

**5. Comment #2 of reviewer #2  (for the second review of the MS)**

*… The argument seems to be that it is too spatially coarse to represent a field, but have sufficient spatial resolution to detect grazing or cutting. I personally do not understand this. The choice is apparently driven by a better temporal resolution (10-days) than Sentinel-2, but the combined Sentinel-2 instruments actually have a shorter revisit time than this, so the only advantage appears to be that the GCLS data are gap-filled. But (a) won't the gap filling itself reduce the ability of the data to represent grazing/cutting? and (b) why not gap fill the Sentinel-2 LAI data? Can the authors provide a better justification for using both data sets?*

**Editor comment :** *the response to this comment seems to be absent from the revised manuscript. I recommend to include a distilled from of the justification for using CGLS into*

*Section 2.1.4. It would probably help to move the last paragraph of Section 2.1.4 it the beginning of that section and add a few more sentences to that paragraph. I agree with reviewer #2 that Section 2.2.2 is unclear as to how the Sentinel 2 data are used. Maybe it would be helpful to state more clearly which parameters were calibrated based on Sentinel vs CGLS?*

**Authors response :** Along the lines of the editor's suggestions we rearranged sections 2.1.4 (P8) and 2.2.2 (P10). We now describe the EO data used in the study in section 2.1.4. At that point the reader will have all the necessary information about the two EO based LAI time-series that we use i.e. their source, spatial/temporal resolution and processing details. We believe that having the justification for using two EO datasets at the materials section can be confusing as this is mostly a methodological aspect of the study. The revised section 2.2.2 now (1) clarifies *why* we need two EO datasets and (2) describes *how* the two EO datasets are used. We use text and the schematic of Figure 3 (P13) to describe the two EO-based time-series are used in our model-data fusion framework.